# DBellQuant: Breaking the Bell with Double-Bell Transformation for LLMs Post Training Binarization

## Abstract

Large language models (LLMs) demonstrate remarkable performance but face substantial computational and memory challenges that limit their practical deployment. Quantization has emerged as a promising solution; however, its effectiveness is often limited by quantization errors arising from weight distributions that are not quantization-friendly and the presence of activation outliers. To address these challenges, we introduce DBellQuant, an innovative post-training quantization (PTQ) framework that achieves nearly 1-bit weight compression and 6-bit activation quantization with minimal performance degradation. DBellQuant uses learnable transformation to map single-bell weight distribution to dual-bell distribution to reduce binarization error and smooth activations using inverse transformation. DBellQuant sets a new state-of-the-art by preserving superior model performance under aggressive weight and activation quantization. For example, on the Wikitext2 dataset, DBellQuant achieves a perplexity of **14.39** on LLaMA2-13B with nearly 1-bit weight and 6-bit activation quantization, significantly outperforming BiLLM's 21.35 without activation quantization, underscoring its potential in compressing LLMs for real-world edge applications.

## 1 Introduction

In recent years, the rapid advancement of large language models (LLMs) demonstrate exceptional performance in a variety of complex tasks that involve natural language understanding and generation (Achiam et al., 2023; Dubey et al., 2024). However, these models often comprise hundreds of billions of parameters, posing significant challenges for their deployment in real-world edge applications because of the substantial computational and memory requirements (e.g. a 70B model requires around 150GB GPU memory), resulting in huge operational costs and unacceptable inference latency.

Quantization is a compelling route to LLM compression, and weight binarization is especially attractive for sharply reducing model size (Wang et al., 2024; Yu et al., 2024). Recent PTQ methods– e.g., PB-LLM and BiLLM– mitigate binarization error with finer-grained treatment of salient weights (Shang et al., 2023; Huang et al., 2024a). However, activation outliers remain a major obstacle to simultaneously quantizing activations (Sun et al., 2024), undercutting memory and latency gains. Scale-redistribution techniques (Xiao et al., 2023; Shao et al., 2023) and Hadamard-based transforms (Ashkboos et al., 2024; Liu et al., 2024; Sun et al., 2025; Lin et al., 2024) help at higher weight precisions, but no PTQ weight binarization method concurrently achieves effective activation quanti-

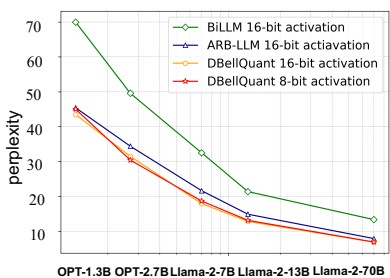

Figure 1: Performance on Wikitext2 dataset. DBellQunat outperforms weight-only quantization method under 8-bit activation setting.

zation. This is because pushing weights to 1-bit increases sensitivity to activation outliers, while existing fixes shift the difficulty back to the weights or require higher-bit activations for accuracy. Quantization-aware training (QAT) approaches such as BitNet can reach 1-bit weights with low-bit activations (Wang et al., 2024), but at the cost of heavy retraining. Thus, there is a need for post-training weight binarization that preserves accuracy while also enabling activation quantization.

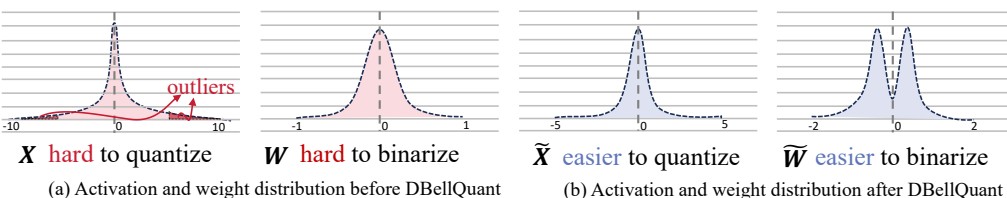

| $X$ hard to quantize | $W$ hard to binarize | $\widetilde{X}$ easier to quantize | $\widetilde{W}$ easier to binarize |

(a) Activation and weight distribution before DBellQuant     (b) Activation and weight distribution after DBellQuant

Figure 2: (a) Before applying DBellQuant, activations exhibit significant outliers, making quantization challenging, while the single-bell-shaped weight distribution hinders binarization. (b) After applying DBellQuant, activations are smoothed with substantially fewer outliers, facilitating easier quantization. Weight distribution is transformed to dual-bell form, which is more conducive to binarization.

To this end, we begin by revisiting the distribution characteristics of activations and weights in LLMs (Fig. 2(a)). We observe that the unimodal nature of weight distributions leads to substantial quantization errors, particularly in the case of low 1-bit quantization. Ideally, a dual-bell-shaped weight distribution (Fig. 2(b)) can effectively reduce binarization errors. In addition, activation distribution suffer from outliers and thus large quantization errors, which demands a narrower distribution. This motivates a key question: *Is it possible to transform weights into a dual-bell shape distribution while simultaneously addressing activation outliers to facilitate both activation quantization and weight binarization?*

Building on these observations, we propose DBellQuant, a novel weight-activation quantization framework for efficient post-training quantization (PTQ). DBellQuant enables activation quantization while achieving near 1-bit weight compression with minimal accuracy loss. The core mechanism is a learnable, equivalence-preserving transformation that maps each layer's weight distribution toward a dual-bell shape; its inverse is applied to the input activations to keep the computation unchanged. We analyze the feasibility of dual-bell transformations for binarization and propose a lightweight algorithm, *Learnable Transformation for Dual-Bell* (LTDB), which initializes the transform and optimizes it with a custom objective that drives weights to cluster around two modes, with early stopping to ensure stability. By doing so, weights originally exhibiting unimodal distributions– challenging for binarization– are mapped into near-symmetric dual-bell distributions (Fig. 2(b)), significantly reducing binarization error. Moreover, as discussed in Sec. 3.4, experiments show that applying the inverse transform to inputs consistently contracts activation ranges and suppresses outliers, making activations more amenable to quantization (Fig. 2(b)).

Experimental results demonstrate that the equivalent transformation strategy of **DBellQuant** significantly reduces the loss caused by weight binarization. For the first time under PTQ conditions, it achieves near 1-bit weight compression while simultaneously compressing activations to 6 bits.Across various LLMs and evaluation metrics, **DBellQuant** consistently achieves state-of-the-art results. For instance, on the Wikitext2 benchmark, we achieves a perplexity of 14.39 on LLaMA2-13B using only 6-bit activations (Fig. 1), significantly surpassing the performance of BiLLM, a method that only quantizes weights, which achieves a perplexity of 21.35.

## 2 RELATED WORK

**Quantization for Large Language Models** The massive parameter size of LLMs poses significant challenges in terms of memory consumption and computational efficiency. Therefore, quantization is crucial to compress these models, reducing resource requirements while preserving performance for practical deployment. LLMs quantization have introduced a variety of innovative techniques to enhance efficiency while maintaining accuracy. Works like GPTQ (Frantar et al., 2022) and OBQ (Frantar et al., 2023b) minimizes reconstruction error by adjusting the remaining unquantized parameters in the block to compensate for the accuracy loss caused by quantization. LLM.int8()(Dettmers et al., 2022a) and ZeroQuant(Yao et al., 2022) improve quantization accuracy by introducing additional grouping labels for customized quantization blocks. Other works like SmoothQuant(Xiao et al., 2023) and OmniQuant (Shao et al., 2023) addresses activation outliers by redistributing scaling factors between weights and activations, migrating the quantization difficulty from activation to weights. Additionally, recent approaches leverage Hadamard transformations to suppress activation outliers (Ashkboos et al., 2024; Liu et al., 2024; Sun et al., 2025), while incoherence processing and non-linear transformation have been proposed for effective low-bit quantization (Chee et al., 2024; Tseng et al., 2024; Zhang et al., 2024). Collectively, these advancements demonstrate that quantiza-

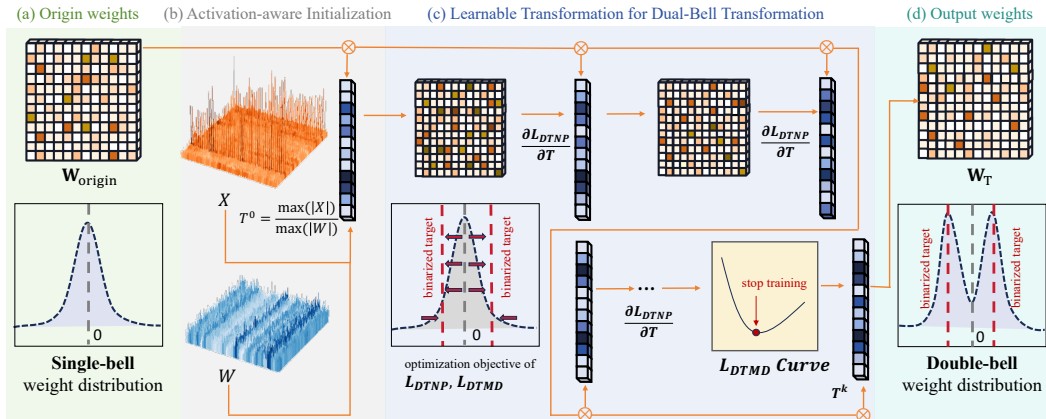

(a) Origin weights    (b) Activation-aware Initialization    (c) Learnable Transformation for Dual-Bell Transformation    (d) Output weights

Figure 3: DBellQuant Framework Overview: (a)First, we can see that the origin weight distribution is single-bell. (b)We utilize Activation-aware initialization to generate origin transformation matrix. (c)We employ the LTDB algorithm for iterative training of the transformation matrix, applying the proposed Dual-Transformation Loss in two ways: for training and as the termination criterion for the training process. (d)The weight distribution after transformation will be double-bell.

tion techniques can be scaled to multi-billion-parameter models, achieving substantial reductions in memory consumption and inference latency without compromising model performance.

**Binary Quantization** Binary quantization, an extreme low-bit quantization technique that reduces model weights and activations to binary values (e.g., -1 and +1 or 0 and 1), has gained significant attention for its ability to drastically cut memory usage and computational complexity, making it ideal for resource-constrained devices and efficient deployment of large-scale models. However, applying binary quantization to LLMs presents substantial challenges due to their sensitivity to precision loss, particularly in attention mechanisms and large embedding layers. BinaryBERT (Bai et al., 2020) explored binary quantization for BERT, proposing selective preservation of critical weights in higher precision to mitigate performance degradation. In another direction, PB-LLM (Shang et al., 2023) introduced a partially-binarized approach for LLMs, retaining a small fraction of salient weights in higher precision while binarizing the rest, enabling extreme low-bit quantization without sacrificing linguistic reasoning capabilities. Recent advancements include structural binarization techniques that leverage novel sparsity forms and standardized importance metrics to selectively binarize and sparsify LLM weights (Dong et al., 2024), as well as strategies like alternating refined binarization and column-group bitmap methods to effectively reduce quantization error and address column deviations (Li et al., 2024). These innovations collectively advance the feasibility of binary quantization for LLMs, pushing the boundaries of efficiency without compromising performance.

## 3 METHOD

We begin by exploring the process of binarization, analyzing and theoretically proving the weight distributions suitable for binarization in Sec. 3.1. Based on our analysis, we propose the Learnable Transformation for Dual-Bell (LTDB) algorithm in Sec. 3.2. After investigating the potential of utilizing a learnable transformation matrix to achieve the objective, we redesigned an efficient learnable transformation along with a reasonable activation-aware initialization method, taking into account training difficulty and task complexity. An overview of the algorithm is included in Fig. 3.

### 3.1 BINARIZATION-FRIENDLY WEIGHT REDISTRIBUTION

By utilizing the sign function, binarization can convert weights in LLMs into binary values. The per-channel binarization and de-binarization process is as follows:

$$\beta = \frac{1}{n}\sum_{i=1}^{n}\boldsymbol{W}_{i,j}, \quad \widetilde{\boldsymbol{W}} = \text{Sign}(\boldsymbol{W} - \beta), \quad \alpha = \frac{1}{n}\sum_{i=1}^{n}|\boldsymbol{W}_{i,j} - \beta_j| \tag{1}$$

$$\text{Sign}(\boldsymbol{W}_{i,j}) = \begin{cases} +1, & \text{if } \boldsymbol{W}_{i,j} > 0, \\ -1, & \text{if } \boldsymbol{W}_{i,j} \leq 0, \end{cases}, \quad \boldsymbol{W}_{deq} = \widetilde{\boldsymbol{W}} \cdot \alpha + \beta \tag{2}$$

where $\beta$ is the shifting factor and $\alpha$ is the scaling factor for binarization. Previous studies (Huang et al., 2024b) have shown that neural network weights exhibit structured distributions along the channel dimension, with certain channels being more salient. The overall weight distribution typically follows a quasi-Gaussian pattern (Dettmers et al., 2022b), as does the channel-wise distribution (Fig. 7). Binarizing such weight matrices introduces significant quantization errors, which can severely degrade model performance.

In LLM binarization, a dual-bell distribution is theoretically more advantageous than a single-bell distribution due to its natural separation into two distinct clusters, which aligns well with binary quantization levels (e.g., -1 and 1), thereby minimizing quantization error. In contrast, single-bell distributions, concentrated around a single peak, often cause significant overlap when mapped to binary values, reducing representation accuracy (see Appendix A.16 for detailed analysis). However, LLM weight distributions typically exhibit single-bell characteristics, and conventional PTQ methods fail to effectively transform them for binarization. While QAT can reshape weight distributions into a dual-bell form through its learning objectives (Wang et al., 2023), it requires substantial computational resources and prolonged training. To address this, we propose a more efficient PTQ method that rapidly converts single-bell distributions into dual-bell ones, optimizing binary quantization without the need for resource-intensive retraining.

## 3.2 Learnable Transformation for Dual-Bell Quantization

**Learnable Transformation with Auxilary Matrix**  As mentioned before, double-bell distributions are advantageous for binarization. However, the key question lies in how to transform a weight matrix that originally follows a single-bell distribution into a double-bell one and ensures the computational results remain unchanged. In this section, we first explore the feasibility of achieving such a transformation through the application of an auxiliary matrix:

**Theorem 1.** *Let $\boldsymbol{W} \in \mathbb{R}^{n \times m}$ be a weight matrix where each channel $\boldsymbol{w}_i$ (for $i \in \{1, 2, \ldots, n\}$) is sampled from a single-bell Gaussian distribution $\boldsymbol{w}_i \sim \mathcal{N}(\mu_i, \sigma_i^2)$. There exists a learnable matrix $T \in \mathbb{R}^{m \times m}$, such that the channels of the transformed matrix $\boldsymbol{W}' = \boldsymbol{W}T$ follow a double-bell distribution, specifically a mixture of two Gaussians:*

$$\boldsymbol{w}_i' \sim \pi \mathcal{N}(\mu_1, \sigma_1^2) + (1 - \pi)\mathcal{N}(\mu_2, \sigma_2^2),$$

*where $\pi \in (0, 1)$ is the mixing coefficient, and $\mu_1, \mu_2, \sigma_1^2, \sigma_2^2$ are parameters of the doubel-bell distribution. More detailed proof is shown in Appendix. A.15.*

Theorem 1 demonstrates that transforming weight distributions from single-bell to double-bell can be achieved by introducing an auxiliary matrix $T$. However, this approach presents several significant challenges. First, in LLMs, weight matrices typically have extremely high dimensionalities, such as $(4096, 4096)$, meaning that the auxiliary matrix $T$ would also be of similarly large dimensions, making it computationally expensive and difficult to learn. Second, to maintain computational consistency, it is necessary to simultaneously apply $T^{-1}$ to the activations, which raises a critical issue regarding the invertibility of $T$. Ensuring strict invertibility introduces additional constraints and complex design steps, further complicating the process. Third, even if $T$ is strictly invertible, it remains uncertain whether this design effectively facilitates activation quantization, as there are no explicit mechanisms in the current approach to optimize activation quantization. These limitations highlight the need for a more efficient and robust design to address the computational and practical challenges associated with auxiliary matrix-based transformations.

**Learnable Equivalent Transformation**  To address these challenges, we propose a simpler and more efficient method for achieving the transformation. In this approach, the matrix $T \in \mathbb{R}^{1 \times C_{\text{in}}}$ to be learned is reduced to a $1 \times C_{\text{in}}$ matrix. Compared to the matrix introduced above, this significantly reduces the dimensionality of $T$ to compared to its original size, making it substantially easier to learn.

Furthermore, this approach allows for straightforward transformations to ensure computational consistency without introducing additional complexity as follows:

$$\mathbf{Y} = \mathbf{X} * \boldsymbol{W} = \mathbf{X} * (T^{-1} * T) * \boldsymbol{W} = \left( \mathbf{X} \odot T^{-1} \right) * (T \odot \boldsymbol{W}) \tag{3}$$

where $X \in \mathbb{R}^{N \times C_{\text{in}}}$ is the input matrix, $N$ is the token length and $C_{\text{in}}$ is the input channel size. $w \in \mathbb{R}^{C_{\text{in}} \times C_{\text{out}}}$ is the weight matrix, where $C_{\text{out}}$ is the output channel size. $\odot$ denotes elementwise multiplication. Moreover, this equivalent transformation matrix $T$ will be directly fused into the LayerNorm weights and the corresponding linear weights, without introducing any additional parameters. However, directly solving this matrix is highly challenging. To address this, we propose a learnable approach to train and derive the matrix effectively.

Here we introduce the way to initialize the learnable transformation matrix $T$ using the following equation:

$$T_j = \frac{\max(|\mathbf{X}_j|)^{\epsilon}}{\max(|\boldsymbol{W}_j|)^{1-\epsilon}} \tag{4}$$

where $\epsilon$ is a hyperparameter. This initialization strategy provides significant advantages for both weight binarization and activation smoothing. For weight quantization, specifically, when $\max(|\boldsymbol{W}_j|)$ is particularly small, it indicates that the absolute value of weights are relatively small, resulting in a large value of $\frac{1}{\max(|\boldsymbol{W}_j|)}$ which corresponds to scaling up these smaller weights. Conversely, when $\max(|\boldsymbol{W}_j|)$ is particularly large, it reflects larger absolute value weights, which lead to a smaller value of $\frac{1}{\max(|\boldsymbol{W}_j|)}$, which will scale down the larger weights. All values can be shifted closer to two central points through these two processes, and it will reduce the quantization error shown in Appendix. A.16. Regarding activation quantization, the initialization explicitly accounts for outliers in the activation matrix, making it inherently activation-aware. This ensures that even without further optimization of activation quantization during subsequent training, it supports near 1-bit weight quantization while effectively reducing activations to a low bit-width.

### 3.3 DUAL-TRANSFORMATION OPTIMIZING OBJECTIVES

**Dual-Target Minimum Deviation Loss**    Our learning objective is to encourage all weight values to move closer to the two mean centers calculated by Eq. 2, ultimately forming a double-bell distribution with these two values as its peaks. During the binarization process, we denote these two points as $m_1$ and $m_2$ respectively. The simple way to set loss function is as follows:

$$\mathcal{L}_{\text{DTMD}} = \frac{\lambda_{\text{DTMD}}}{n} \sum_{i=1}^{n} \min \left( |\boldsymbol{W} * T_i - m_{1,i}|, |\boldsymbol{W} * T_i - m_{2,i}| \right) \tag{5}$$

where $\lambda_{\text{DTMD}}$ represents the coefficient of $\mathcal{L}_{\text{DTMD}}$. However, employing this type of loss function to train the transformation matrix introduces an issue. Specifically, we observed that the transformation matrix tends to shrink progressively during training, contrary to the intended effect of scaling up the originally smaller absolute values. This unintended behavior results in a significant problem: it effectively shifts the quantization challenge from the weights to the activations.

**Dual-Target Normalized Proportional Loss**    As discussed before, DTMD alone is not enough to train a better weights distribution for binarization. So we introduce a new loss function as follows:

$$\mathcal{L}_{\text{DTNP}} = \frac{\lambda_{DTNP}}{n} \sum_{i=1}^{n} \begin{cases} \frac{|\boldsymbol{W} * T_i - \text{m}_{1,i}|}{|\text{m}_{1,i}|}, & \text{if } |\boldsymbol{W} * T_i - \text{m}_{1,i}| < |\boldsymbol{W} * T_i - \text{m}_{2,i}| \\ \frac{|\boldsymbol{W} * T_i - \text{m}_{2,i}|}{|\text{m}_{2,i}|}, & \text{otherwise.} \end{cases} \tag{6}$$

where $\lambda_{DTNP}$ represents the coefficient of $\text{Loss}_{\text{rel}}$. By leveraging the dual-target normalized proportional objective, the transformation matrix can be effectively trained to meet the desired behavior, scaling down larger absolute values and scaling up smaller absolute values to approach a double-bell-shaped distribution. Since our target values have been transformed into $\frac{|\boldsymbol{W} * T_i - \text{m}_i|}{|\text{m}_i|}$, the final convergence values might not align with our desired results. Furthermore, we propose an early

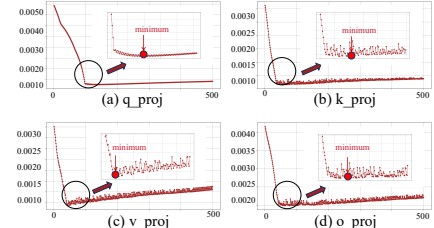

Figure 4: Dual-Target Minimum Deviation Loss value over iterations across different layers.

stopping mechanism to prevent the function from converging to an undesired solution that deviates from our intended objective. We observe that by using this loss to train, DTMD drops quickly first and then slowly grow as shown in Fig. 4. Therefore, we introduce an early stop mechanism and DTMD is utilized as a condition for stopping the training.

### 3.4 IMPACT OF THE INVERSE OF LEARNABLE TRANSFORMATION MATRIX ON ACTIVATION SMOOTHING

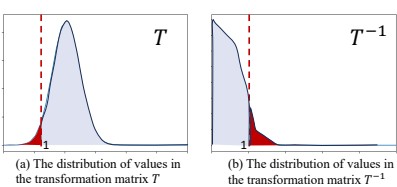

(a) The distribution of values in the transformation matrix $T$

(b) The distribution of values in the transformation matrix $T^{-1}$

Figure 5: Visualization of distribution of values in $T$ and $T^{-1}$ of Llama2-7B.

Through the use of DTNP for training, we understand that the transformation matrix $T$ drives all values in the weights closer to the two mean centers calculated by Eq. 2. Prior research has shown that the distribution of weights tends to approximate a quasi-Gaussian distribution, with the majority of values being extremely small and close to zero, while only a very small fraction exhibit relatively large absolute values (Dettmers et al., 2022b). Theoretically, this implies that $T$ will contain many values greater than 1 to amplify the numerous near-zero absolute values, bringing them closer to the mean centers. At the same time, very few values of $T$ will be less than 1. In fact, when we visualize $T$ after training as shown in Fig. 5, we observe that less than 5% of its values are below 1. Consequently, the corresponding $T^{-1}$, which is multiplied with activations, has over 95% of its values below 1. This effectively narrows the distribution of activation, making quantization easier. Visualization results can be seen in Appendix. A.13 and we also provide quantitative analysis of activation outliers in Appendix. A.14. As a result, this approach smooths activation quantization.

### 3.5 ALGORITHM

The process of Learnable Transformation for Dual-Bell Transformation Algorithm is shown in Algorithm. 1, which can be seen in Appendix. A.4 and can adjust the full-precision weight matrix $W$ using an activation-aware initialization transformation matrix $T$ over $N$ epochs. The algorithm iteratively minimizes dual-target loss functions to guide $W$ toward a dual-bell distribution while employing an early stopping mechanism based on the DTMD.

## 4 EXPERIMENTS

### 4.1 SETTINGS

All experimental procedures were executed utilizing the PyTorch (Paszke et al., 2019) framework in conjunction with the Huggingface library (Paszke et al., 2019). Models with parameters smaller than 8B are running on a single NVIDIA A30 GPU equipped with 24GB of memory, others are running on a single NVIDIA A100 GPU equipped with 80GB of memory. Consistent with methodologies outlined by Frantar et al. (Frantar et al., 2023a) and Huang et al. (Huang et al., 2024a), a calibration dataset comprising 128 samples sourced from the C4 collection (Raffel et al., 2020) was employed.

**Models and Datasets** Comprehensive evaluations were carried out across several large language model families, including LLaMA, LLaMA-2, and LLaMA-3 (Touvron et al., 2023) and the OPT series (Zhang et al., 2022). The efficacy of the developed DBellQuant was assessed by calculating the perplexity of the models' generated text on standard benchmarks: WikiText2 (Merity et al., 2017), and a subset of the C4 data (Raffel et al., 2020). Furthermore, the models' performance was evaluated based on accuracy across five zero-shot question-answering tasks: ARC-c (Clark et al., 2018), ARC-e (Clark et al., 2018), Hellaswag (Zellers et al., 2019), PIQA (Bisk et al., 2020), and Winogrande (Sakaguchi et al., 2020).

**Comparison Methods** The primary benchmark for comparison for our DBellQuant approach is BiLLM (Huang et al., 2024a), which represents the current baseline PTQ technique for binary large language models. Additionally, we include other contemporary PTQ algorithms in our comparison, namely Round-to-Nearest (RTN), GPTQ (Frantar et al., 2023a), PB-LLM (Shang et al., 2023) and the current state-of-the-art PTQ technique ARB-LLM (Li et al., 2024).

Table 1: Perplexity of RTN, GPTQ, PB-LLM, BiLLM, ARB-LLM$_X$ and our methods on **OPT** and **LLaMA** family. The columns represent the perplexity results on **WikiText2** datasets with different model sizes.

| Method | Activation Bits | OPT-1.3B | OPT-2.7B | OPT-6.7B | LLaMA-1-7B | LLaMA-2-7B | LLaMA-2-13B | LLaMA-2-70B |
|---|---|---|---|---|---|---|---|---|
| Full Precision | 16 | 14.62 | 12.47 | 10.86 | 5.68 | 5.47 | 4.88 | 3.32 |
| RTN | 16 | 17165.72 | 36516.69 | 11550.91 | 168388.00 | 157058.34 | 47902.32 | 160389.91 |
| GPTQ | 16 | 14844.73 | 14114.58 | 10622.81 | 267001.72 | 115905.67 | 9387.80 | 14219.35 |
| PB-LLM | 16 | 265.52 | 124.35 | 105.16 | 102.36 | 69.20 | 151.09 | 28.37 |
| BiLLM | 16 | 69.97 | 49.55 | 35.36 | 35.04 | 32.48 | 21.35 | 13.32 |
| ARB-LLM$_X$ | 16 | 45.40 | 34.37 | 20.07 | 21.81 | 21.61 | 14.86 | 7.88 |
| **DBellQuant** | 16 | **43.42** | **31.47** | **18.89** | **15.34** | **17.91** | **12.79** | **6.84** |
| BiLLM | 8 | 88.95 | 68.60 | 166.46 | 40.13 | 33.23 | 22.55 | 14.72 |
| **DBellQuant** | 8 | **44.98** | **30.39** | **18.88** | **14.74** | **18.65** | **13.11** | **6.88** |
| BiLLM | 6 | 9537 | 18405.85 | 28123.58 | 71.65 | 42.41 | 30.20 | 18.65 |
| **DBellQuant** | 6 | **61.50** | **47.33** | **21.12** | **16.66** | **21.69** | **14.39** | **7.56** |

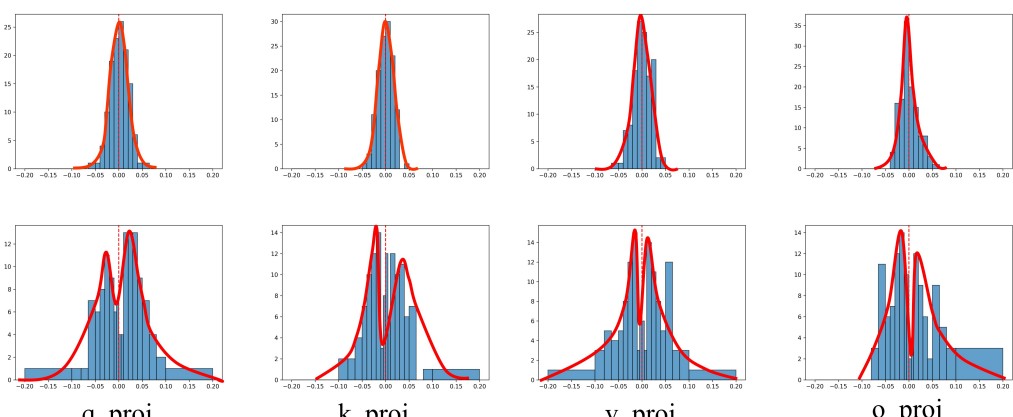

q_proj      k_proj      v_proj      o_proj

Figure 6: **Top:** Visualization of single-bell weights distribution from different blocks of different layers before applying DBellQuant. **Bottom:** Visualization of dual-bell weights distribution from different blocks of different layers after applying DBellQuant.

## 4.2 MAIN RESULTS

LLMs across different activation quantization bit-widths and model sizes, deploying DBellQuant with a block size of 128. As shown in Tab. 1, we compare the WikiText2 perplexity of the OPT and Llama families across different model sizes. The results demonstrate that DBellQuant significantly outperforms the state-of-the-art ARB-LLM$_X$ when only quantizing weights, achieving up to a 42.18% reduction in perplexity. Moreover, when activations are quantized to lower bit-widths like 6-bit, DBellQuant achieves up to a 76.66% reduction in perplexity for the LLaMA family compared to BiLLM. It is noteworthy that for OPT family, the model outputs under the BiLLM methods have already collapsed when quantizing the activation to 6 bit, whereas DBellQuant still maintains reasonable linguistic output capabilities. In terms of average accuracy on QA datasets, DBellQuant also significantly surpasses previous methods and increase the average accuracy up to 42.48%, as detailed in Tab. 2. Here, we only compare with ARB-LLM$_X$ because we share the same quantization format. Additionally, Fig. 7 visualizes the transformation of weight distributions across different layers, clearly illustrating the shift from a single-bell to a double-bell distribution. These results highlight the effectiveness of DBellQuant in enabling robust low-bit quantization while preserving model performance. More results can be seen in Appendix. A.12.

## 4.3 ABLATION EXPERIMENTS

**Effectiveness of LTDB algorithm** To validate the effectiveness of our advanced LTDB algorithm, we compare them with the vanilla initialization without LTDB. We can observe that under different bitwidths of activation quantization, the models utilizing the proposed LTDB method consistently

Table 2: Accuracy of PIQA, ARC-e, ARC-c, HellaSwag, Winogrande and average accuracy of all datasets with BiLLM and our methods on OPT and LLaMA family.

| Model | Method | Activation Bits | PIQA | ARC-e | Arc-c | HellaSwag | Winogrande | Avg. |
|-------|--------|-----------------|------|-------|-------|-----------|------------|------|
| OPT-6.7B | - | 16 | 76.33 | 65.61 | 30.55 | 50.51 | 65.35 | 57.67 |
| | BiLLM | 16 | 59.63 | 36.83 | 17.06 | 30.14 | 51.30 | 38.99 |
| | ARB-LLM$_X$ | 16 | 69.75 | 55.47 | 24.32 | 37.78 | 58.64 | 49.19 |
| | **DBellQuant** | 16 | **70.29** | **55.76** | **24.72** | **37.81** | **58.71** | **49.46** |
| | BiLLM | 8 | 54.30 | 31.02 | 20.05 | 26.66 | 50.90 | 36.59 |
| | **DBellQuant** | 8 | **69.10** | **54.63** | **24.91** | **38.19** | **57.38** | **48.84** |
| | BiLLM | 6 | 53.43 | 20.08 | 20.22 | 25.80 | 47.67 | 33.44 |
| | **DBellQuant** | 6 | **68.12** | **52.44** | **23.38** | **37.04** | **57.30** | **47.65** |
| LLaMA-1-7B | - | 16 | 78.40 | 67.34 | 38.14 | 56.45 | 67.01 | 61.46 |
| | BiLLM | 16 | 61.92 | 38.93 | 21.58 | 32.78 | 53.67 | 41.77 |
| | ARB-LLM$_X$ | 16 | 67.23 | 49.13 | 23.98 | 39.21 | 58.69 | 47.64 |
| | **DBellQuant** | 16 | **67.74** | **49.37** | **24.23** | **39.55** | **58.80** | **47.94** |
| | BiLLM | 8 | 61.86 | 37.88 | 21.76 | 32.09 | 51.62 | 41.04 |
| | **DBellQuant** | 8 | **67.41** | **47.31** | **26.19** | **38.89** | **58.80** | **47.72** |
| | BiLLM | 6 | 57.45 | 31.06 | 20.65 | 29.78 | 53.04 | 38.40 |
| | **DBellQuant** | 6 | **65.29** | **46.46** | **25.77** | **37.28** | **54.85** | **45.94** |
| LLaMA-2-7B | - | 16 | 78.40 | 69.28 | 40.02 | 56.69 | 67.25 | 62.32 |
| | BiLLM | 16 | 60.34 | 36.87 | 21.59 | 30.24 | 51.62 | 40.13 |
| | ARB-LLM$_X$ | 16 | 63.33 | 42.35 | 21.25 | 34.51 | 55.56 | 43.4 |
| | **DBellQuant** | 16 | **63.98** | **42.85** | **23.89** | **34.82** | **56.27** | **44.36** |
| | BiLLM | 8 | 59.74 | 36.95 | 21.42 | 30.96 | 53.75 | 40.56 |
| | **DBellQuant** | 8 | **62.56** | **42.42** | **23.03** | **34.44** | **54.38** | **43.37** |
| | BiLLM | 6 | 56.81 | 28.32 | 20.05 | 29.33 | 52.17 | 37.33 |
| | **DBellQuant** | 6 | **61.10** | **37.5** | **22.18** | **31.94** | **53.85** | **41.32** |
| LLaMA-3-8B | - | 16 | 79.65 | 80.09 | 50.51 | 60.18 | 72.77 | 68.64 |
| | BiLLM | 16 | 57.51 | 33.75 | 18.52 | 31.63 | 53.12 | 38.90 |
| | ARB-LLM$_X$ | 16 | 63.73 | 42.75 | 21.79 | 34.41 | 55.89 | 43.71 |
| | **DBellQuant** | 16 | **64.15** | **44.11** | **22.02** | **34.73** | **56.36** | **44.27** |
| | BiLLM | 8 | 60.55 | 37.96 | 18.34 | 32.60 | 51.78 | 40.24 |
| | **DBellQuant** | 8 | **61.70** | **40.95** | **18.40** | **32.95** | **55.09** | **41.82** |
| | BiLLM | 6 | 56.09 | 33.92 | 16.72 | 31.75 | 51.85 | 38.06 |
| | **DBellQuant** | 6 | **58.00** | **37.63** | **18.77** | **31.83** | **51.92** | **39.64** |

| Bit-width | LTDB Algorithm | WikiText2 ↓ | C4 ↓ |
|-----------|----------------|-------------|------|
| 16 | ✗ | 24.41 | 27.69 |
| 16 | ✓ | **17.91** | **21.83** |
| 8 | ✗ | 27.39 | 30.91 |
| 8 | ✓ | **18.65** | **23.80** |
| 6 | ✗ | 32.74 | 42.48 |
| 6 | ✓ | **21.69** | **30.24** |

| Method | Activation Bits | Block Size | WikiText2 ↓ | C4 ↓ |
|--------|-----------------|------------|-------------|------|
| BiLLM | 16 | 64 | 20.12 | 24.46 |
| DBellQuant | 8 | 64 | 13.67 | 16.99 |
| DBellQuant | 6 | 64 | 15.65 | 19.71 |
| BiLLM | 16 | 128 | 32.48 | 40.52 |
| DBellQuant | 8 | 128 | 18.65 | 23.02 |
| DBellQuant | 6 | 128 | 21.69 | 25.91 |
| BiLLM | 16 | 256 | 43.69 | 43.21 |
| DBellQuant | 8 | 256 | 22.34 | 23.37 |
| DBellQuant | 6 | 256 | 24.68 | 28.52 |

Table 3: Performance w/o LTDB Algorithm.   Table 4: Performance of different block size.

**(a)** Performance of different loss function.

| Method | Loss Function | Activation Bits | WikiText2 ↓ | C4 ↓ |
|--------|---------------|-----------------|-------------|------|
| DBellQuant | L2 | 8 | 18.90 | 24.11 |
| **DBellQuant** | **L1** | 8 | **18.65** | **23.02** |
| DBellQuant | L2 | 6 | 22.26 | 26.41 |
| **DBellQuant** | **L1** | 6 | **21.69** | **25.91** |

**(b)** Performance of different $\epsilon$

| Method | $\epsilon$ | WikiText2 ↓ | C4 ↓ |
|--------|------------|-------------|------|
| DBellQuant | 0.75 | 19.57 | 22.92 |
| DBellQuant | 0.8 | 18.66 | 22.81 |
| **DBellQuant** | **0.85** | **17.91** | **21.83** |
| DBellQuant | 0.9 | 18.29 | 22.52 |

Table 5: Ablation study on LLaMA-2-7B, results are measured by perplexity.

outperform those relying on vanilla initialization as shown in Tab. 3. This demonstrates that the proposed LTDB method can effectively reduce quantization errors.

| Method | 2 bit | 1.1 bit |
|--------|-------|---------|
| QuaRot+RTN | inf | inf |
| QuaRot+GPTQ | 22.07 | Inf |
| DBellQuant | - | 17.91 |

Table 6: Llama-2-7B results.

| Method | 2 bit | 1.1 bit |
|--------|-------|---------|
| QuaRot+RTN | inf | inf |
| QuaRot+GPTQ | 10.41 | Inf |
| DBellQuant | - | 12.79 |

Table 7: Llama-2-13B results.

| Method | wikitext2 | C4 |
|--------|-----------|-----|
| DBellQuant | 17.91 | 21.83 |
| DBellQuant+QuaRot | 62.37 | 76.59 |

Table 8: Combination results

| Method | wikitext2 | C4 |
|--------|-----------|-----|
| BiLLM | 38.42 | 40.81 |
| DBellQuant | 30.47 | 35.02 |

Table 9: Qwen-2-7B results.

| Method | wikitext2 | C4 |
|--------|-----------|-----|
| BiLLM | 41.74 | 53.07 |
| DBellQuant | 33.47 | 43.18 |

Table 10: Qwen-2.5-7B results.

| Loss Function | wikitext2 | C4 |
|---------------|-----------|-----|
| DTMD+DTNP | 17.91 | 21.83 |
| only DTNP | 28.74 | 31.42 |

Table 11: Loss Function results

**Optimization Objective**    In the definition of DTNP, we adopted L1 loss as the objective function, as it consistently outperformed L2 loss in our experiments. Additionally, we introduced the hyperparameter $\epsilon$ during the activation-aware initialization process and found that all tested values surpassed prior algorithms, with $\epsilon = 0.85$ achieving the best performance for LLaMA-2-7B as shown in Tab. 5. Detailed parameter settings are provided in the Appendix. A.5.

**Impact of Block Size**    We evaluated the impact of block size on DBellQuant's quantization performance using block sizes ranging from 64 to 256 columns, as shown in Tab. 4. Similar to other PTQ methods, smaller block sizes achieve lower perplexity but increase quantization diversity and weighting overhead. In previous experiments, we used a block size of 128, as it balances bit-width and quantization performance effectively. Notably, our method consistently outperforms baseline approaches across all block sizes, highlighting its robustness and generalizability.

**Activation Bit-width Comparisons**    We compared model performance across different activation bit-widths, as shown in Fig. 8. When activations are quantized to 8 bits, perplexity remains nearly unchanged or even improves, demonstrating our method's ability to mitigate activation quantization challenges. Even with lower-bit quantization, such as 6-bit, the performance remains largely intact, with minimal perplexity increases. Notably, for large-scale models like LLaMA-2 13B and 70B, perplexity degradation is negligible, underscoring the effectiveness of our approach for models with substantial parameters.

**Effectiveness of DTMD and DTNP Loss**    With DTMD loss, T tends to shrink toward zero, leading to a "loss hack" effect that prevents the desired transformation. To address this, we introduce the DTNP loss to ensure a smooth transition. However, using DTNP alone creates a new issue, as its convergence direction can conflict with DTMD. As shown in Tab. 11, using DTNP alone drives the model toward a direction misaligned with DTMD, resulting in degraded performance.

**Comparison and Combination with Rotation-based Methods**    Rotation-based PTQ methods, such as QuaRot with Hadamard-based transforms, effectively reduce outliers in weights and activations. We compared low-bit quantization using these methods alone and in combination with our approach. As shown in Tab.6 and Tab.7, rotation-based PTQ methods experience significant performance degradation when weight precision is reduced below 4-bit. Therefore, we focus on comparisons with state-of-the-art methods for near-1-bit weight quantization. Additionally, we evaluated the combination of DBellQuant and Hadamard-based transforms, such as QuaRot, on the Llama-2-7B model, but this combination performed worse than DBellQuant alone, as shown in Tab. 8. A possible explanation is that DBellQuant transforms the original unimodal distribution into a binarization-friendly bimodal distribution, while Hadamard-based transforms disrupt this structure, negatively impacting quantization performance.

**Implementation on Qwen Family Models**    The Qwen model has recently attracted significant attention for its strong performance. We conducted preliminary evaluations of quantization methods on Qwen-2-7B and Qwen-2.5-7B, with results presented in Tab.9 and Tab.10. These results show that DBellQuant outperforms BiLLM on both Qwen models, demonstrating its adaptability and effectiveness not only on OPT and LLaMA models but also on more advanced LLMs.

| Method | MathQA | LogiQA2 |
|---|---|---|
| fp16 | $44.96 \pm 0.91$ | $29.58 \pm 1.15$ |
| GPTQ 2bit | $20.51 \pm 0.82$ | $14.65 \pm 1.53$ |
| BiLLM 1.1bit | $24.99 \pm 0.79$ | $20.42 \pm 1.87$ |
| DBellQuant 1.1bit | $27.65 \pm 0.75$ | $22.78 \pm 1.48$ |

Table 12: DeepSeek-R1-Distill-Qwen-7B results.

| Method | TextVQA | ChartQA | MME-Per |
|---|---|---|---|
| fp16 | 84.9 | 87.3 | 1695 |
| GPTQ 2bit | 0 | 0 | 319 |
| BiLLM 1.1bit | 26.3 | 3.7 | 638 |
| DBellQuant 1.1bit | 28.4 | 5.2 | 742 |

Table 13: Qwen-2.5-VL-7B results.

| Method | Model | mins ↓ | WikiText2 ↓ |
|---|---|---|---|
| BiLLM | LLaMA-1-7B | 35 | 35.04 |
| ARB-LLM$_X$ | LLaMA-1-7B | 88 | 21.81 |
| DBellQuant | LLaMA-1-7B | 47 | 15.34 |
| BiLLM | LLaMA-2-7B | 37 | 32.48 |
| DBellQuant | LLaMA-2-7B | 49 | 17.91 |

Table 14: Training time comparison.

| Method | Bit-width ↓ | Model Size ↓ | Perplexity ↓ |
|---|---|---|---|
| FP16 | 16 | 13.5GB | 5.47 |
| RTN | 2 | 2.31GB | 1e5 |
| GPTQ | 2 | 2.31GB | 60.45 |
| PB-LLM | 1.7 | 2.08GB | 69.20 |
| BiLLM | 1.08 | 1.98GB | 32.48 |
| DBellQuant | 1.15 | 2.04GB | 17.91 |

Table 15: Bit-width and model size comparison.

**Implementation on Reasoning and Multi-modal Models**  For the reasoning model, we used DeepSeek-R1-Distill-Qwen-7B and evaluated it on MathQA and LogiQA2. As shown in Tab.12, while performance drops noticeably when quantized to near 1-bit, our method still outperforms GPTQ 2-bit and BiLLM, highlighting its advantage in ultra-low-bit quantization. For multi-modal large models, we evaluated Qwen-2.5-VL-7B on TextVQA, ChartQA, and MME, with results presented in Tab.13. Although performance decreases significantly at near 1-bit quantization, our method consistently outperforms GPTQ 2-bit, where some tasks produce invalid outputs (e.g., gibberish), leading to zero scores. Additionally, our approach surpasses BiLLM, further demonstrating its effectiveness in ultra-low-bit quantization.

## 4.4 TIME AND MEMORY ANALYSIS

**Time Comparison**  We clarify that the additional training step introduced by LDTB incurs minimal overhead while yielding substantial performance gains. Specifically, compared to BiLLM, optimizing the learnable T matrix for the LLaMA2-7B model requires only 12 additional minutes on an A100 80G GPU as shown in Tab. 14, a negligible cost given the performance improvements achieved, reducing the perplexity on WikiText2 from 32.48 to 17.91.

| Model | 4096×4096 | 4096×11008 | 11008×4096 |
|---|---|---|---|
| FP16 | 0.79463 | 1.73942 | 1.82653 |
| BiLLM | 0.36842 | 0.38744 | 0.43906 |
| ARB-LLMX | 0.33180 | 0.35539 | 0.36792 |
| **DBellQuant** | **0.27694** | **0.30085** | **0.31860** |

Table 16: Actual inference speed comparison on various linear layers in LLaMA-2-7B.

Moreover, compared to ARB-LLM$_X$, our approach not only achieves lower perplexity but also reduces training time. More results can be seen in Appendix. A.9. The actual inference speedup is shown in Tab. 16 and more details are in Appendix. A.11.

**Memory Comparison**  We include a detailed comparison of bitwidth and model size across different settings. Our method reduces the model size to approximately 1/7 to 1/6 of the original FP16 model, with only a minimal increase in perplexity. Compared to other ultra-low-bit PTQ methods such as BiLLM and PB-LLM, our approach achieves comparable bitwidth and model size, while yielding significantly lower perplexity as shown in Tab. 15. More results can be seen in Appendix. A.10.

## 5 CONCLUSION

We propose DBellQuant, an efficient PTQ method enabling simultaneous weight binarization and activation quantization for LLMs. By leveraging weight distributions suited for binarization, we design the Learnable Transformation for Dual-Bell algorithm. This includes two customized loss functions and an early stopping mechanism to achieve the dual-bell transformation. This transformation enhances activation quantization, enabling near 1-bit weight compression and 6-bit activation quantization with minimal performance loss—achieving this milestone for the first time in a PTQ framework. Experiments on open-source LLMs demonstrate that DBellQuant significantly advances the performance of SOTA binary PTQ methods.

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

# A  APPENDIX

## A.1  USE OF LARGE LANGUAGE MODELS

In preparing this manuscript, we exclusively used large language models—GPT-5 (OpenAI, 2025)—to refine grammar, flow, and tone at the sentence and paragraph levels. These tools were not used to generate ideas, design experiments, or draw conclusions. All technical content, methodologies, and interpretations were independently authored, rigorously verified, and approved by the authors. To prevent factual inaccuracies or citation errors, every model-edited sentence was reviewed by the authors, and all references were meticulously cross-checked with their original sources. The authors take full responsibility for the accuracy and integrity of this work.

## A.2  LIMITATIONS

Currently, our work only supports quantizing activations to 6 bits. When attempting to quantize activations to lower bit-widths, the model collapses, resulting in a significant drop in performance. However, some recent studies have demonstrated the ability to quantize activations to 4 bits while maintaining competitive model performance. Inspired by these advancements, we aim to adopt similar approaches to further optimize our models. Specifically, we seek to not only binarize weights but also quantize activations to even lower bit-widths, enabling easier deployment and faster computation.

## A.3  BROADER IMPACTS

Our work demonstrates the feasibility of simultaneously binarizing weights and quantizing activations to 6 bits in LLMs while maintaining competitive performance. This approach significantly reduces the computational and memory overhead associated with LLM deployment, making them more accessible for resource-constrained environments. By enabling efficient inference, our method contributes to the democratization of advanced AI technologies, reducing the environmental impact of large-scale model deployment. Furthermore, it opens new avenues for research in ultra-low-bit quantization, fostering innovation in model efficiency and scalability.

## A.4  ALGORITHM

The detailed algorithm is shown in Algorithm. 1.

---

**Algorithm 1** Learnable Transformation for Dual-Bell Transformation (LTDB)

---

1: **function** LTDB($\boldsymbol{W}, \mathbf{T}, N$)
2:    **Input:** $\boldsymbol{W} \in \mathbb{R}^{n \times m}$ - a full-precision weight matrix.
3:        $\mathbf{T} \in \mathbb{R}^{1 \times m}$ - an activation-aware initialization transformation matrix.
4:        $N$ - the total number of epochs.
5:    **Output:** $\widetilde{\boldsymbol{W}} \in \mathbb{R}^{n \times m}$ - the transformed weight matrix.
6:    **for** iter $= 1, 2, \ldots, N$ **do**
7:        $\widetilde{\boldsymbol{W}} \leftarrow \mathbf{T} \odot \boldsymbol{W}$                    ▷ Perform element-wise multiplication
8:        $\mathcal{L}_{\text{DTMD}}, \mathcal{L}_{\text{DTNP}} \leftarrow \text{LossFunc}(\widetilde{\boldsymbol{W}})$       ▷ Compute the two dual-target losses for $\widetilde{\boldsymbol{W}}$
9:        $\mathcal{L}_{\text{DTNP}}.backward()$       ▷ Use Dual-Target Normalized Proportional Loss for training
10:       **if** $\mathcal{L}_{\text{DTMD}} > \mathcal{L}_{\text{DTMD-Minimum}}$ **then**
11:           **break**              ▷ Stop training based on Dual-Target Minimum Deviation Loss
12:       **end if**
13:       $\mathcal{L}_{\text{DTMD-Minimum}} \leftarrow \mathcal{L}_{\text{DTMD}}$
14:    **end for**
15:    **return** $\widetilde{\boldsymbol{W}}$
16: **end function**

---

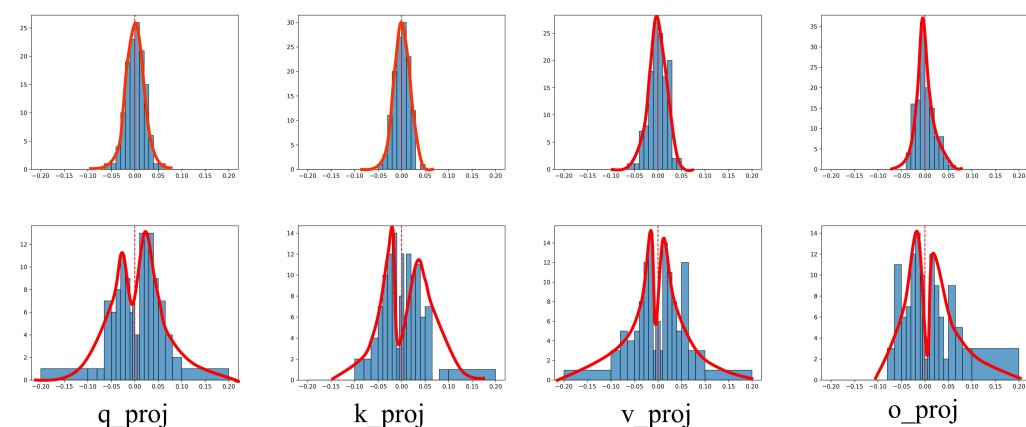

q_proj          k_proj          v_proj          o_proj

Figure 7: **Top:** Visualization of single-bell weights distribution from different blocks of different layers before applying DBellQuant. **Bottom:** Visualization of dual-bell weights distribution from different blocks of different layers after applying DBellQuant.

## A.5    HYPERPARAMETER SETTING IN TRAINING

Our method involves a few key hyperparameters, including the loss coefficient , the number of training epochs, and the learning rate. These were initially determined by validating on the LLaMA2-7B model, where we used a loss coefficient of 100, 200 epochs, and a learning rate of 0.01.

For the smooth parameter $\epsilon$ , we followed the strategy of SmoothQuant, testing values of 0.75, 0.80, 0.85, and 0.90. The differences in performance were marginal, with 0.85 yielding the best results, and thus selected as the default. The stopping criterion is based on LTDB loss mentioned in Section. 3.3.

Importantly, we found that these hyperparameters generalize well across different architectures and scales—including both the LLaMA and OPT model families—without requiring re-tuning. This demonstrates that our method is robust and not sensitive to hyperparameter choices.

## A.6    VISUALIZATION OF THE TRANSFORMATION OF WEIGHT DISTRIBUTIONS ACROSS DIFFERENT LAYERS

The visualization of weights distribution before and after our proposed method DBellQuant is shown in Fig. 7.

## A.7    DETAILED APPLICATION OF TRANSFORMATION

To preserve computational consistency and avoid introducing additional parameters, we apply the transformation $T$ between the LayerNorm and the q/k/v projection layers, as well as between the LayerNorm and the up/gate projection layers. This placement mirrors that of SmoothQuant, ensuring the transformed inputs are compatible with quantization while maintaining the model's architecture.

For $o\_proj$, although it is theoretically possible to insert a transformation $T$ between $v\_proj$ and $o\_proj$ to complete the symmetry, we found that this leads to conflicting transformations on $v\_proj$, degrading performance. Therefore, we do not apply any transformation to $o\_proj$ in practice.

Similarly, for $down\_proj$, the presence of activation functions (e.g., GeLU) makes it non-trivial to apply transformations without disrupting the computation graph or introducing inconsistencies. As a result, we do not insert transformations in this part either.

But we need to clarify that both $o\_proj$ and $down\_proj$'s weight are binarized in DBellQuant. To clarify, for these two layers, $o\_proj$ and $down\_proj$, we did not learn a T-transformation to modify their weights. However, we still applied the original binarization in BiLLM to process these two layers, making our approach a w1a6 method.

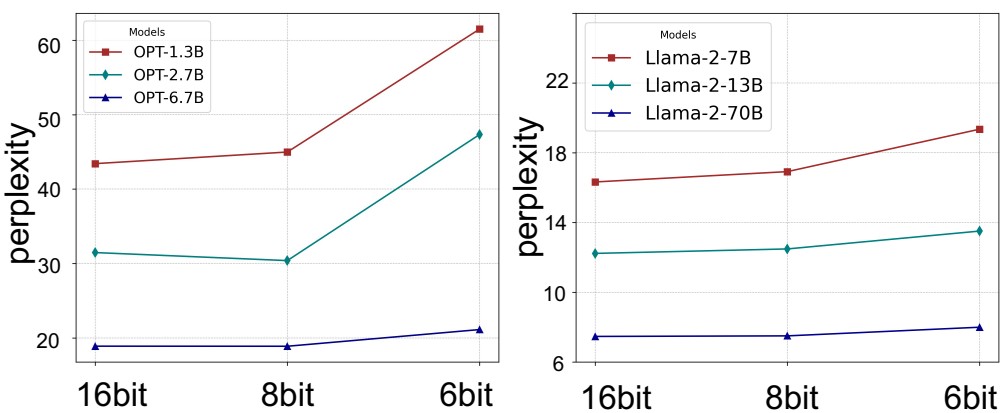

Figure 8: Performance of different activation bit-widths.

### A.8 PERFORMANCE OF DIFFERENT ACTIVATION BIT-WIDTHS

The performance of different activation bit-widths across different models is shown in Fig. 8.

### A.9 RESULTS ABOUT TRAINING OF DIFFERENT METHODS

Detailed training time and perplexity of different methods across various models are shown in Tab. 17.

Table 17: Traininig Time and Performance Comparison

|          | OPT-1.3B | | OPT-2.3B | | OPT-6.7B | | LLaMA-1-7B | | LLaMA-2-7B | |
|----------|------|-------|------|-------|------|-------|------|-------|------|-------|
|          | mins | ppl   | mins | ppl   | mins | ppl   | mins | ppl   | mins | ppl   |
| BiLLM    | 7    | 69.97 | 13   | 49.55 | 34   | 35.36 | 35   | 35.04 | 37   | 32.48 |
| ARB-LLMX | –    | –     | –    | –     | –    | –     | 88   | 21.81 | –    | –     |
| DBellQuant | 9  | 43.42 | 17   | 31.47 | 45   | 18.89 | 47   | 15.34 | 49   | 17.91 |

### A.10 RESULTS ABOUT BIT-WIDTH, MODEL SIZE AND PERPLEXITY OF DIFFERENT METHODS

Detailed bit-width, model size and perplexity of different methods across various models are shown in Tab. 18, Tab. 19 and Tab. 20.

Table 18: Bit-width, Model Size and Performance Comparison for OPT-6.7B

| Methods | bit-width | model size | ppl |
|---------|-----------|------------|-----|
| FP16    | 16        | 12.5GB     | 10.86 |
| RTN     | 2         | 2.15GB     | 2e4 |
| GPTQ    | 2         | 2.15GB     | 50.19 |
| PB-LLM  | 1.7       | 1.95GB     | 105.16 |
| BiLLM   | 1.11      | 1.84GB     | 35.36 |
| DBellQuant | 1.18   | 1.89GB     | 18.89 |

### A.11 RESULTS ABOUT COMPUTATION SPEEDUP

To quantify computational speedup, we follow the benchmarking strategy of ARB-LLM and utilize the BitBLAS codebase, which supports mixed-precision GEMM operations for low-bit weights. We benchmark the latency (in milliseconds) of linear layers in LLaMA2-7B using input sequences of length 2048. The results are shown in Tab. 21. Our findings are as follows:

Table 19: Bit-width, Model Size and Performance Comparison for LLaMA-1-7B

| Methods | bit-width | model size | ppl |
|---|---|---|---|
| FP16 | 16 | 13.5GB | 5.68 |
| RTN | 2 | 2.25GB | 1e5 |
| GPTQ | 2 | 2.25GB | 152.31 |
| PB-LLM | 1.7 | 2.05GB | 102.36 |
| BiLLM | 1.09 | 1.97GB | 35.04 |
| DBellQuant | 1.15 | 2.02GB | 15.34 |

Table 20: Bit-width, Model Size and Performance Comparison for LLaMA-2-7B

| Methods | bit-width | model size | ppl |
|---|---|---|---|
| FP16 | 16 | 13.5GB | 5.47 |
| RTN | 2 | 2.31GB | 1e5 |
| GPTQ | 2 | 2.31GB | 60.45 |
| PB-LLM | 1.7 | 2.08GB | 69.20 |
| BiLLM | 1.08 | 1.98GB | 32.48 |
| DBellQuant | 1.15 | 2.04GB | 17.91 |

1)Weight binarization significantly reduces inference latency compared to FP16 models, thanks to faster memory access and bitwise computation

2)DBellQuant supports 8-bit activation quantization, enabling efficient INT8 inference. This not only accelerates runtime but also yields better perplexity compared to prior binary weight methods like BiLLM and ARB-LLMX.

Table 21: Computation Speed Comparison

| Model | 4096×4096 | 4096×11008 | 11008×4096 |
|---|---|---|---|
| FP16 | 0.79463 | 1.73942 | 1.82653 |
| BiLLM | 0.36842 | 0.38744 | 0.43906 |
| ARB-LLMX | 0.33180 | 0.35539 | 0.36792 |
| **DBellQuant** | **0.27694** | **0.30085** | **0.31860** |

## A.12 PERPLEXITY ON C4 DATASET

As shown in Tab.22, we compare the perplexity of the OPT and LLaMA families across different model sizes on C4 dataset.

## A.13 VISUALIZATION OF DISTRIBUTION OF ACTIVATION BEFORE AND AFTER DBELLQUANT

In this section, we present the changes in the distribution of activation values before and after applying DBellQuant as shown in Fig.9. It is evident that the extreme values in the activation have been significantly reduced by a factor of 5 to 10; for instance, the maximum value decreases from approximately 3 to around 0.4. Previous studies have highlighted that one of the primary challenges in low-bit quantization of activations lies in the presence of large outliers, which expand the activation range and, consequently, amplify quantization errors. By applying DBellQuant, the activation range is effectively compressed from [-3, 3] to [-0.4, 0.4], dramatically alleviating the difficulty of quantization. This reduction in range establishes highly favorable conditions for further exploration of lower-bit quantization, such as 8-bit or even 6-bit implementations.

Table 22: Perplexity of RTN, GPTQ, PB-LLM, BiLLM, ARB-LLM$_X$ and our methods on **OPT** and **LLaMA** family. The columns represent the perplexity results on **C4** datasets with different model sizes.

| Method | Activation Bits | OPT-1.3B | OPT-2.7B | OPT-6.7B | LLaMA-1-7B | LLaMA-2-7B | LLaMA-2-13B | LLaMA-2-70B |
|---|---|---|---|---|---|---|---|---|
| Full Precision | 16 | 16.07 | 14.34 | 12.71 | 7.34 | 7.26 | 6.73 | 5.71 |
| RTN | 16 | 9999.56 | 23492.89 | 9617.07 | 194607.78 | 115058.76 | 46250.21 | 314504.09 |
| GPTQ | 16 | 6364.65 | 6703.36 | 5576.82 | 186229.5 | 67954.04 | 19303.51 | 13036.32 |
| PB-LLM | 16 | 168.12 | 222.15 | 104.78 | 76.63 | 80.69 | 184.67 | NAN |
| BiLLM | 16 | 64.14 | 44.77 | 42.13 | 46.96 | 39.38 | 25.87 | 17.30 |
| ARB-LLM$_X$ | 16 | 47.60 | 34.97 | 22.54 | 22.73 | 28.02 | 19.82 | 11.85 |
| **DBellQuant** | 16 | **42.57** | **32.89** | **21.78** | **17.60** | **21.83** | **15.14** | **9.49** |
| BiLLM | 8 | 74.56 | 61.99 | 40.91 | 47.13 | 40.91 | 21.45 | 17.72 |
| **DBellQuant** | 8 | **44.60** | **32.52** | **21.56** | **18.16** | **23.80** | **15.56** | **9.61** |
| BiLLM | 6 | 7348 | 13445.21 | 63.41 | 61.65 | 63.41 | 37.66 | 19.43 |
| **DBellQuant** | 6 | **57.14** | **45.24** | **23.12** | **19.80** | **30.24** | **17.84** | **10.12** |

## A.14 ANALYSIS OF RELATIVE ERROR OF THE ACTIVATION

To further validate our method, we conducted experiments demonstrating its effectiveness in reducing the relative error of activations. Specifically, we randomly sampled 128 data points from the C4 dataset and extracted the $q_{\text{proj}}$ inputs from the LLaMA2-7B model using both BiLLM (without applying the inverse transform on activations) and our proposed DBellQuant (with the inverse transform applied).

We evaluated the relative error using two metrics: the Z-score, as you suggested, and the relative deviation error. The formulas are as follows:

- **Z-score:**

$$Z = \frac{x - \mu}{\sigma}$$

   where $x$ is the value, $\mu$ is the mean, and $\sigma$ is the standard deviation.

- **Relative Deviation Error:**

$$\text{Relative Deviation Error} = \frac{x - u}{u}$$

   where $u$ is the mean value of the corresponding row.

These metrics allowed us to comprehensively assess the relative error and validate the robustness of our proposed method. The computation results are shown in Tab. 23.

| | Z-score | Relative Deviation Error |
|---|---|---|
| **BiLLM** | 0.1584 | 33.78 |
| **DBellQuant** | 0.1401 | 30.72 |

Table 23: Comparison of Z-score and Relative Deviation Error between BiLLM and DBellQuant.

Our method demonstrates its effectiveness not only in reducing the absolute error of activations but also in minimizing outliers from a relative distance perspective, as verified by DBellQuant.

## A.15 PROOF FOR THEOREM. 1

*Proof.* **Problem Setup:** By assumption, the rows of the original weight matrix $\mathbf{W}$ are sampled independently from Gaussian distributions:

$$\mathbf{w}_i \sim \mathcal{N}(\mu_i, \sigma_i^2), \quad \text{where } \mu_i \in \mathbb{R} \text{ and } \sigma_i > 0 \text{ for all } i.$$

We aim to learn a transformation matrix $\mathbf{T} \in \mathbb{R}^{m \times m}$ such that the rows of the resulting matrix $\mathbf{W}' = \mathbf{W}\mathbf{T}$ follow a bimodal distribution.

**Learnable Transformation Definition:** The transformed matrix is defined as:

$$\mathbf{W}' = \mathbf{W}\mathbf{T},$$

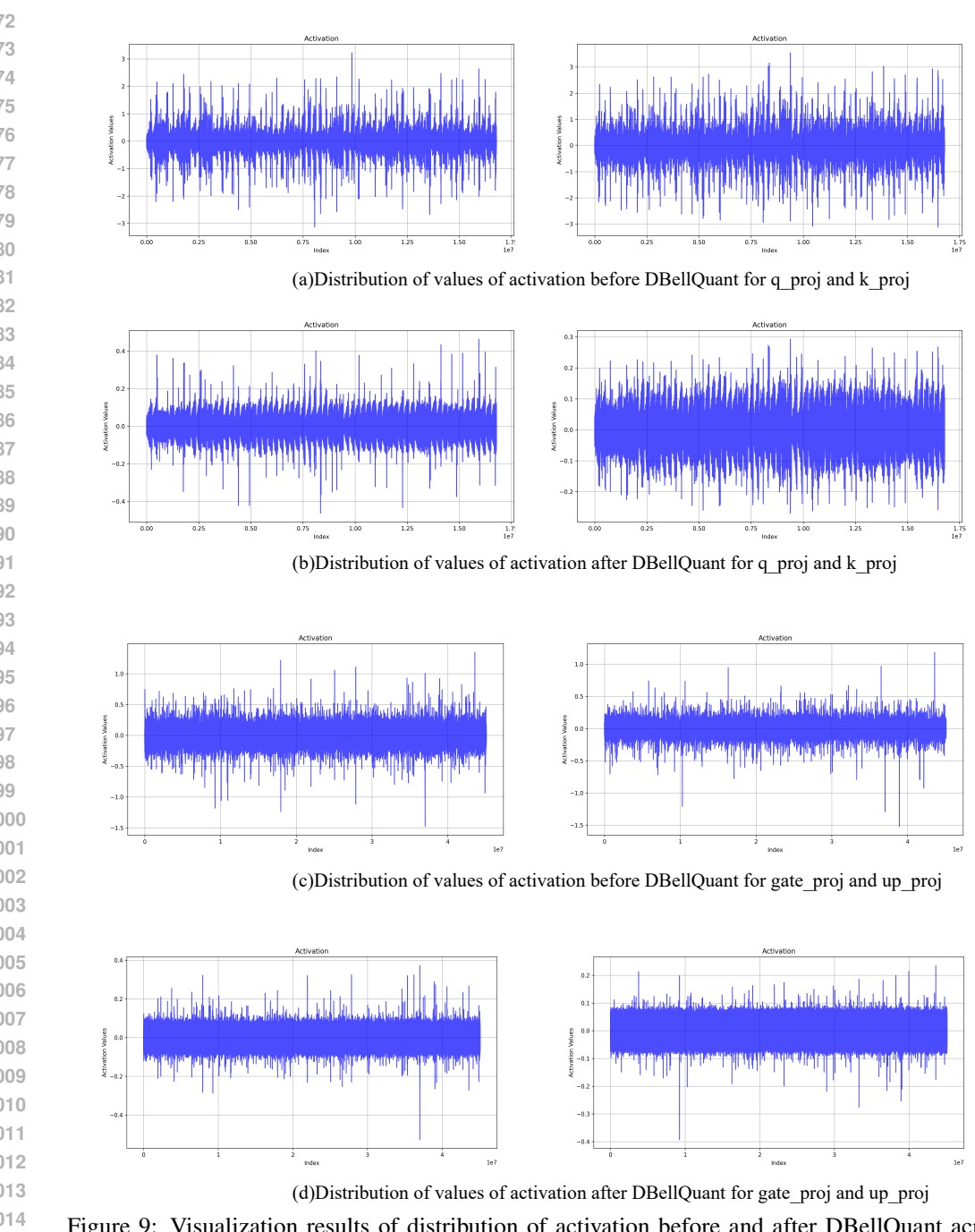

(a)Distribution of values of activation before DBellQuant for q_proj and k_proj

(b)Distribution of values of activation after DBellQuant for q_proj and k_proj

(c)Distribution of values of activation before DBellQuant for gate_proj and up_proj

(d)Distribution of values of activation after DBellQuant for gate_proj and up_proj

Figure 9: Visualization results of distribution of activation before and after DBellQuant across different blocks.

where $\mathbf{T}$ is a learnable matrix that modulates the distribution of each row $\mathbf{w}'_i$ of $\mathbf{W}'$. Since the rows of $\mathbf{W}$ are Gaussian-distributed, the linear transformation by $\mathbf{T}$ initially results in a new Gaussian distribution for each row:

$$\mathbf{w}'_i \sim \mathcal{N}(\mu'_i, \sigma'^2_i),$$

where $\mu'_i = \mu_i \mathbf{T}$ and $\sigma'^2_i = \mathbf{T}^\top \Sigma_i \mathbf{T}$, with $\Sigma_i = \mathrm{diag}(\sigma^2_i)$ being the covariance of $\mathbf{w}_i$.

**Inducing a Bimodal Distribution:** To map the Gaussian-distributed rows $\mathbf{w}'_i$ into a bimodal distribution, we note that a bimodal distribution can be expressed as a Gaussian mixture model:

$$g(x) = \pi \mathcal{N}(x; \mu_1, \sigma^2_1) + (1 - \pi)\mathcal{N}(x; \mu_2, \sigma^2_2),$$

where $\pi \in (0, 1)$ is the mixing coefficient, and $\mu_1, \mu_2, \sigma_1^2, \sigma_2^2$ are the parameters of the mixture components. To achieve this, $\mathbf{T}$ is learned to ensure that the linear transformation $\mathbf{WT}$ reshapes the original Gaussian distribution into a mixture of two Gaussians.

**Parameter Optimization:** The learnable matrix $\mathbf{T}$ is optimized using a loss function $L$ that minimizes the Kullback-Leibler (KL) divergence between the empirical distribution of the rows of $\mathbf{W}'$ and the target bimodal distribution:

$$L = \mathrm{KL}\left(p(\mathbf{w}_i') \,\|\, \pi\mathcal{N}(\mu_1, \sigma_1^2) + (1 - \pi)\mathcal{N}(\mu_2, \sigma_2^2)\right).$$

The optimization process adjusts the entries of $\mathbf{T}$ to align the transformed rows $\mathbf{w}_i'$ with the desired bimodal distribution.

**Conclusion:** The existence of such a learnable matrix $\mathbf{T}$ ensures that the rows of the transformed matrix $\mathbf{W}' = \mathbf{WT}$ can follow a bimodal distribution. This completes the proof. $\qquad\square$

### A.16 ANALYSIS OF THE REASONS DOUBLE-BELL DISTRIBUTION MORE SUITABLE FOR BINARIZATION COMPARED TO A SINGLE-BELL DISTRIBUTION.

Directly proving that a dual-bell distribution is more suitable for binarization compared to a single-bell distribution can be challenging, as it requires setting numerous additional conditions. However, this problem becomes significantly simpler when approached from the perspective of value adjustments. By reducing the magnitude of larger absolute values and increasing smaller absolute values in a bimodal distribution, all values can be shifted closer to two central points, effectively creating a double-bell-like distribution. We can demonstrate that this approach reduces the quantization loss introduced by binarization, thereby supporting the suitability of double-bell distributions for this purpose.

**Theorem 2.** *Given an input calibration activation $x \in \mathbb{R}^{n \times 1}$ and a weight vector $\boldsymbol{w} \in \mathbb{R}^{n \times 1}$, where $w_i$ is extracted from the weight matrix $\boldsymbol{W} \in \mathbb{R}^{n \times n}$ along a specific channel, we define the weight vector $\boldsymbol{w}$ as the union of two sets: - A set of several **outliers** with large absolute values, denoted as $U_o = \{o_1^*, o_2^*, \ldots, o_k^*\}$, where $|o_i^*| \gg 0$ for $i \in \{1, \ldots, k\}$; - A set of **normal values** with small absolute values, denoted as $U_n = \{n_1, n_2, \ldots, n_{n-k}\}$, where $|n_j| \approx 0$ for $j \in \{1, \ldots, n - k\}$. $\boldsymbol{w} = U_o \cup U_n$. We now define a new weight vector $\boldsymbol{w}_{new}$ as follows:*

- *$\boldsymbol{w}_{new} = [n_1, n_2, \ldots, \gamma o_1^*, \gamma o_2^*, \ldots, \gamma o_k^*, \ldots, n_{n-k}]$, where $\gamma \in (\frac{1}{2}, 1)$.*

- *Alternatively, $\boldsymbol{w}_{new} = [\eta n_1, \eta n_2, \ldots, o_1^*, o_2^*, \ldots, o_k^*, \ldots, \eta n_{n-k}]$, where $\eta \in (1, 2)$.*

*Then, the quantization error induced by $\boldsymbol{w}_{new}$, defined as $\|x \cdot \boldsymbol{w} - x \cdot binarized(\boldsymbol{w}_{new})\|$, is strictly smaller than the original quantization error $\|x \cdot \boldsymbol{w} - x \cdot binarized(\boldsymbol{w})\|$ in both cases.*

*Proof.* A.16.1 SCALING UP SMALL VALUES REDUCES QUANTIZATION ERROR

Consider the scenario where the input vector is $X = [2, 2, 2, \ldots, 2]_n$. Assume the weights in a single channel, $W$, are given by $[\alpha_1, \alpha_2, \ldots, \alpha_k, \beta_1, \beta_2, \beta_3, \ldots, \beta_{n-k}]$, where $[\alpha_i \in U_1]$ is the set of values with very large absolute magnitudes, satisfying $\sum_{i=1}^{k} \alpha_i = A$ and $[\beta_i \in U_2]$ are values with magnitudes close to zero, satisfying $\sum_{i=1}^{n-k} \beta_i = 0$. $W = U_1 \cup U_2$. This structure is common in practice, as weight distributions in neural networks often exhibit a few dominant values and many small ones. As we observe, the distribution of weights along the channel dimension is mostly not symmetric around zero. Instead, it tends to be biased, with the majority leaning either towards positive or negative values. Consequently, the extreme values are predominantly either entirely positive or entirely negative, so we assume $\alpha_i > 0$.

The product of the input $X$ and the weight vector $W$ is:

$$X \cdot W^T = 2(\alpha_1 + \alpha_2 + \cdots + \alpha_k + \beta_1 + \beta_2 + \cdots + \beta_{n-k}) = 2(\alpha_1 + \alpha_2 + \cdots + \alpha_k) = 2A \quad (7)$$

since the sum of the $\beta_i$ is zero.

According to the quantization function, the mean value $M$ is:

$$M = \frac{\alpha_1 + \alpha_2 + \cdots + \alpha_k + \beta_1 + \beta_2 + \cdots + \beta_{n-k}}{n} = \frac{A}{n} \quad (8)$$

The absolute mean value, $AbsMean$, is defined as:

$$AbsMean = \frac{|\alpha_1 - \frac{A}{n}| + |\alpha_2 - \frac{A}{n}| + \cdots + |\alpha_k - \frac{A}{n}| + |\beta_1 - \frac{A}{n}| + |\beta_2 - \frac{A}{n}| + \cdots + |\beta_{n-k} - \frac{A}{n}|}{n}$$

$$(9)$$

Given that $[\beta_1, \beta_2, \ldots, \beta_{n-1}]$ are all very small in magnitude compared to $\frac{\alpha}{n}$, we can simplify the above as:

$$
\begin{aligned}
AbsMean &= \frac{(\alpha_1 - \frac{A}{n}) + (\alpha_2 - \frac{A}{n}) + \cdots + (\alpha_k - \frac{A}{n}) + (\frac{A}{n} - \beta_1) + (\frac{A}{n} - \beta_2) + \cdots + (\frac{A}{n} - \beta_{n-k})}{n} \\
&= \frac{A + (n - 2k)\frac{A}{n} - (\beta_1 + \beta_2 + \cdots + \beta_{n-k})}{n} \\
&= \frac{A + (n - 2k)\frac{A}{n}}{n} \\
&= \frac{2(A - \frac{kA}{n})}{n}
\end{aligned}
$$

$$(10)$$

Here, the sum of the $\beta_i$ vanishes due to their zero sum constraint.

Therefore, the dequantized value for $\alpha$ is:

$$AbsMean + M = \frac{2(A - \frac{2kA}{n})}{n} + \frac{A}{n} \tag{11}$$

and for each $\beta_i$:

$$-AbsMean + M = -\frac{2(A - \frac{2kA}{n})}{n} + \frac{A}{n} \tag{12}$$

To reduce the quantization error associated with the small-magnitude weights, we can scale them up by a factor $m > 1$, while correspondingly scaling down the input. Specifically, we multiply each $\beta_i$ by $m$ and divide the associated input elements by $m$. The new input becomes $X_{new} = [2, 2, \ldots, 2, \frac{2}{m}, \frac{2}{m}, \ldots, \frac{2}{m}]_n$, and the new weights are $W_{new} = [\alpha_1, \alpha_2, \ldots, \alpha_k, m\beta_1, m\beta_2, \ldots, m\beta_{n-k}]$.

The output remains unchanged:

$$X_{new} \cdot W_{new}^T = 2(\alpha_1 + \alpha_2 + \cdots + \alpha_k) + \frac{2}{m} \cdot m\beta_1 + \frac{2}{m} \cdot m\beta_2 + \cdots + \frac{2}{m} \cdot m\beta_{n-k} = 2A \tag{13}$$

This invariance is crucial: the scaling operation does not affect the original computation, but it can impact the quantization error.

For the new weights, the mean value is:

$$M_{new} = \frac{\alpha_1 + \alpha_2 + \cdots + \alpha_k + m\beta_1 + m\beta_2 + \cdots + m\beta_{n-k}}{n} = \frac{A + m(\beta_1 + \beta_2 + \cdots + \beta_{n-k})}{n} = \frac{A}{n} \tag{14}$$

since the sum of the $\beta_i$ is zero.

The new absolute mean value is:

$$AbsMean_{new} = \frac{|\alpha_1 - \frac{A}{n}| + |\alpha_2 - \frac{A}{n}| + \cdots + |\alpha_k - \frac{A}{n}| + |m\beta_1 - \frac{\alpha}{n}| + |m\beta_2 - \frac{\alpha}{n}| + \cdots + |m\beta_{n-k} - \frac{\alpha}{n}|}{n}$$

$$(15)$$

If $m$ is chosen such that $\frac{\alpha}{n}$ remains larger than all $m\beta_i$, the simplification proceeds as before:

$$
\begin{aligned}
AbsMean_{new} &= \frac{(\alpha_1 - \frac{A}{n}) + (\alpha_2 - \frac{A}{n}) + \cdots + (\alpha_k - \frac{A}{n}) + (\frac{A}{n} - m\beta_1) + (\frac{A}{n} - m\beta_2) + \cdots + (\frac{A}{n} - m\beta_{n-k})}{n} \\
&= \frac{A + (n - 2k)\frac{A}{n} - m(\beta_1 + \beta_2 + \cdots + \beta_{n-k})}{n} \\
&= \frac{A + (n - 2k)\frac{A}{n}}{n} \\
&= \frac{2(A - \frac{kA}{n})}{n}
\end{aligned}
\tag{16}
$$

Thus, $AbsMean_{new}$ and $M_{new}$ are identical to $AbsMean$ and $M$, and the dequantized values are unchanged. This demonstrates that scaling up the small weights does not affect the mean or absolute mean, but it can improve the quantization error, as we analyze next.

Let us now analyze the quantization error. The original output is $2\alpha$. For convenience, let $N = (n - k)(-AbsMean + M)$. The quantized output is a sum of the dequantized values for all weights.

$A > 0$ Both before and after scaling, the quantization output contains the term:

$$
2k(AbsMean + M) = 2k \left( \frac{2(A - \frac{kA}{n})}{n} + \frac{A}{n} \right)
\tag{17}
$$

For $n$ typically greater than 100 and $k \ll n$ it holds that:

$$
0 < 2k(AbsMean + M) < 2A
\tag{18}
$$

This is because the quantized value is always less than the original due to the averaging effect.

For the term $-AbsMean + M$:

$$
\begin{aligned}
-AbsMean + M &= -\frac{2(A - \frac{kA}{n})}{n} + \frac{A}{n} \\
&= -\frac{A}{n} + \frac{2kA}{n^2} \\
&= \frac{A}{n} \left( \frac{2k}{n} - 1 \right)
\end{aligned}
\tag{19}
$$

Since $n > 100$ ,$A > 0$ and $k \ll n$ this value is negative and its magnitude is small.

The quantization output before scaling is $2k(AbsMean + M) + 2N$, and after scaling is $2k(AbsMean + M) + N$. Because $N < 0$ and $2(AbsMean + M) < 2A$, we have:

$$
2k(AbsMean + M) + 2N < 2k(AbsMean + M) + N < 2A
\tag{20}
$$

This shows that scaling up the small weights reduces the quantization error, as the quantized output moves closer to the original value.

if $\alpha_i < 0$, the proof is similar.

In summary, scaling up the small weights (and correspondingly scaling down the input) does not change the original computation, but it systematically reduces the quantization error by making the quantized output more faithful to the original.

### A.16.2  SCALING DOWN LARGE VALUES REDUCES QUANTIZATION ERROR

Now, let us consider the scenario where we scale down the large-magnitude weight. Let the input $X = [1, 1, 1, \ldots, 1]_n$, and the weights $W = [m\alpha_1, m\alpha_2, ..., m\alpha_k, \beta_1, \beta_2, \beta_3, \ldots, \beta_{n-k}]$, where $\alpha_1, \alpha_2, \ldots, \alpha_k$ are large values, satisfying $\sum_{i=1}^{k} \alpha_i = A$ and $\alpha_i > 0$, $[\beta_1, \ldots, \beta_{n-k}]$ are small, and $\sum_{i=1}^{n-k} \beta_i = 0$.

The output is:

$$
X \cdot W^T = m(\alpha_1 + \alpha_2 + \cdots + \alpha_k + \beta_1 + \beta_2 + \cdots + \beta_{n-k} = mA
\tag{21}
$$

The mean value is:

$$M = \frac{m\alpha_1 + m\alpha_2 + \cdots + m\alpha_k + \beta_1 + \beta_2 + \cdots + \beta_{n-k}}{n} = \frac{mA}{n} \tag{22}$$

The absolute mean is:

$$AbsMean = \frac{|m\alpha_1 - \frac{mA}{n}| + |m\alpha_2 - \frac{mA}{n}| + \cdots + |m\alpha_k - \frac{mA}{n}| + |\beta_1 - \frac{mA}{n}| + |\beta_2 - \frac{mA}{n}| + \cdots + |\beta_{n-k} - \frac{mA}{n}|}{n} \tag{23}$$

Since $\frac{mA}{n}$ is much larger than the $\beta_i$, we can simplify:

$$AbsMean = \frac{(m\alpha_1 - \frac{mA}{n}) + (m\alpha_2 - \frac{mA}{n}) + \cdots + (m\alpha_k - \frac{mA}{n}) + (\frac{mA}{n} - \beta_1) + (\frac{mA}{n} - \beta_2) + \cdots + (\frac{mA}{n} - \beta_{n-k})}{n}$$

$$= \frac{m(\alpha_1 + \alpha_2 + \cdots + \alpha_k) + (n - 2k)\frac{mA}{n} - (\beta_1 + \beta_2 + \cdots + \beta_{n-1})}{n}$$

$$= \frac{mA + (n - 2k)\frac{mA}{n}}{n}$$

$$= \frac{2m(A - \frac{kA}{n})}{n} \tag{24}$$

The dequantized value for $m\alpha$ is:

$$AbsMean + M = \frac{2m(A - \frac{kA}{n})}{n} + \frac{mA}{n} = \frac{3mA - \frac{2mkA}{n}}{n} \tag{25}$$

and for each $\beta_i$:

$$-AbsMean + M = -\frac{2m(A - \frac{A}{n})}{n} + \frac{mA}{n} = \frac{-mA + \frac{2mkA}{n}}{n} \tag{26}$$

To scale down the large value $m\alpha$, we divide it by $m$ ($m > 1$) and multiply the corresponding input element by $m$. The new input is $X_{new} = [m, 1, 1, \ldots, 1]_n$, and the new weights are $W_{new} = [\alpha_1, \alpha_2, \ldots, \alpha_k, \beta_1, \beta_2, \ldots, \beta_{n-k}]$.

The output remains unchanged:

$$X_{new} \cdot W_{new}^T = m(\alpha_1 + \alpha_2 + \cdots + \alpha_k) + \beta_1 + \beta_2 + \cdots + \beta_{n-k} = mA \tag{27}$$

For the new weights, the mean is:

$$M_{new} = \frac{\alpha_1 + \alpha_2 + \cdots + \alpha_k + \beta_1 + \beta_2 + \cdots + \beta_{n-k}}{n} = \frac{A}{n} \tag{28}$$

The new absolute mean is:

$$AbsMean_{new} = \frac{|\alpha_1 - \frac{A}{n}| + |\alpha_2 - \frac{A}{n}| + \cdots + |\alpha_k - \frac{A}{n}| + |\beta_1 - \frac{A}{n}| + |\beta_2 - \frac{A}{n}| + \cdots + |\beta_{n-1} - \frac{A}{n}|}{n} \tag{29}$$

With appropriate $m$, we have:

$$AbsMean_{new} = \frac{(\alpha_1 - \frac{A}{n}) + (\alpha_2 - \frac{A}{n}) + \cdots + (\alpha_k - \frac{A}{n}) + (\frac{\alpha}{n} - \beta_1) + (\frac{\alpha}{n} - \beta_2) + \cdots + (\frac{\alpha}{n} - \beta_{n-k})}{n}$$

$$= \frac{(\alpha_1 + \alpha_2 + + \cdots + \alpha_k) + (n - 2k)\frac{A}{n} - (\beta_1 + \beta_2 + \cdots + \beta_{n-1})}{n}$$

$$= \frac{A + (n - 2k)\frac{A}{n}}{n}$$

$$= \frac{2(A - \frac{kA}{n})}{n} \tag{30}$$

The dequantized value for $\alpha_i$ is:

$$AbsMean_{new} + M_{new} = \frac{2(A - \frac{kA}{n})}{n} + \frac{A}{n} = \frac{3A - \frac{2kA}{n}}{n} \tag{31}$$

and for each $\beta_i$:

$$-AbsMean_{new} + M_{new} = -\frac{2(A - \frac{kA}{n})}{n} + \frac{A}{n} = \frac{-A + \frac{2kA}{n}}{n} \tag{32}$$

Let us now examine the quantization error. The original output is $m\alpha$. The quantization output before scaling is:

$$k(AbsMean + M) + (n - k)(-AbsMean + M)$$
$$= k\frac{3mA - \frac{2mkA}{n}}{n} + (n - k)\frac{-mA + \frac{2mkA}{n}}{n} \tag{33}$$

The quantization output after scaling is:

$$km(AbsMean_{new} + M_{new}) + (n - k)(-AbsMean_{new} + M_{new})$$
$$= km\frac{3A - \frac{2kA}{n}}{n} + (n - k)\frac{-A + \frac{2kA}{n}}{n} \tag{34}$$
$$= k\frac{3mA - \frac{2mkA}{n}}{n} + (n - k)\frac{-A + \frac{2kA}{n}}{n}$$

Because $A > 0$, for $n > 100$, $\alpha > 0$ and $k \ll n$ it holds that:

$$0 < k\frac{3mA - \frac{2mkA}{n}}{n} < mA \tag{35}$$

and

$$\frac{-A + \frac{2kA}{n}}{n} = \frac{A}{n}\left(\frac{2k}{n} - 1\right) < 0 \tag{36}$$

Comparing the quantization outputs, we see:

$$k\frac{3mA - \frac{2mkA}{n}}{n} + m(n - k)\frac{-A + \frac{2kA}{n}}{n}$$
$$< k\frac{3mA - \frac{2mkA}{n}}{n} + (n - k)\frac{-A + \frac{2kA}{n}}{n} \tag{37}$$
$$< mA$$

If $\alpha_i < 0$, the proof is similar.

$$\square$$

