# OpenReview forum: "DBellQuant: Breaking the Bell with Double-Bell Transformation for LLM Post Training Binarization"
_ICLR.cc/2026/Conference — Submitted to ICLR 2026_

### Official Review · Reviewer_ETLQ · 2025-10-15

**Soundness:** 2
**Presentation:** 2
**Contribution:** 3
**Rating:** 4
**Confidence:** 5

**Summary:**

This paper’s key strength lies in pioneering near 1-bit weight and 6-bit activation post-training quantization (PTQ) for large language models (LLMs) with minimal performance loss, aligning with practical deployment needs for low memory and latency. However, it has critical weaknesses—including reproducibility issues from unspecified algorithm details, unclear connections to prior work, unaddressed causal/theoretical gaps in core mechanisms, lack of significant innovation, and minor errors—leading to a "Reject" score, though revision of these issues could warrant reconsideration.

**Strengths:**

1. This work achieves near 1-bit weight quantization combined with 6-bit activation quantization in the Post-Training Quantization (PTQ) scenario for the first time, with minimal performance loss. From a research direction perspective, it aligns with practical deployment needs of Large Language Models (LLMs), such as low memory usage and low latency.

**Weaknesses:**

1. There is a spelling error in Tables 4 and 5: the term "Activation" is misspelled as "Avtivation".

2. The iterative steps of the core LTDB algorithm (Section 3.2) are only provided as simplified pseudocode in Appendix A.4. Critical details such as the optimizer type for gradient updates (e.g., Adam/SGD) and learning rate scheduling strategy are not specified. For the loss function coefficients λ_DTMD and λ_DTNP (Section 3.3), only their existence is mentioned, while specific values are not provided. For instance, Appendix A.5 only vaguely refers to a "loss coefficient of 100" without clarifying which loss term it corresponds to. These omissions severely hinder the reproducibility of the proposed method.

3. The "Related Work" section (Section 2) merely lists existing methods without highlighting the connection between core limitations of prior work and the research gap addressed by this paper. For example, the essential differences between SmoothQuant’s "scaling factor redistribution" and the proposed "T-transformation" are not clearly compared or analyzed.

4. In the method section (Section 3), the description of "how the T-matrix is integrated into LayerNorm weights" (Section 3.2) is only mentioned in a single sentence. No formulas or schematic diagrams are provided to illustrate the integration details, leaving this critical implementation step unclear.

5. The paper only observes that "95% of the values of T⁻¹ are less than 1, thus compressing the activation range" (Section 3.4). However, it fails to explain a key logical question: why the inverse transformation of the T-matrix (designed for weight quantization) can恰好 suppress activation outliers? No causal relationship or theoretical reasoning is provided to support this observation.

6. The reason why the DTMD loss causes T-matrix shrinkage is not analyzed. The paper only states that "DTMD causes T to shrink and needs to be compensated by DTNP" (Section 3.3), but does not explain why the gradient updates of DTMD tend to reduce the values of T (e.g., mathematical derivation of gradient direction). Additionally, no comparative experiments (e.g., using only DTNP vs. combining DTMD+DTNP) are conducted to verify the necessity of using two loss functions.

7. The work lacks significant innovation. Adjusting weight distribution to adapt to quantization is not a new idea—QAT (Quantization-Aware Training) methods (e.g., BitNet) have already shaped weights into a double-bell distribution through training. This paper merely migrates this concept to the PTQ scenario. Moreover, there is no novel design in the "distribution transformation method": the T-transformation is essentially element-wise scaling, which is similar to the "weight-activation scaling factor redistribution" idea of SmoothQuant, with only the addition of learnability.

Based on the above weakness, I would assign a Reject score. I look forward to the authors addressing the aforementioned problems in future revisions, and I would be happy to reconsider and raise my score accordingly.

**Questions:**

See "Weaknesses"

---

> ### Author Response · Authors · 2025-11-25
> **Response to Reviewer ETLQ(1/3)**
>
> Dear Reviewer ETLQ,
>
> Thank you for highlighting and appreciating our contributions in exploring the ultra-low-bit weight quantization in LLM. Below, we will address your questions one by one.
>
> >Q1: There is a spelling error … misspelled as "Avtivation".
>
> Thank you for pointing this out. We acknowledge the typo and will ensure it is corrected in future revisions of the manuscript.
>
> >Q2: The iterative steps … the proposed method.
>
> In the **LTDB algorithm**, we use **stochastic gradient descent (SGD)** as the optimizer **without any learning‑rate scheduling**. The **learning rate is fixed at 0.01** throughout all experiments, which we found to provide stable convergence across models of different sizes.
>
> Regarding the loss formulation in **Section 3.3**, the **loss coefficient mentioned in Appendix A.5 corresponds to  λDTNP**, which is **set to 100** for all experiments. This value was empirically determined to balance convergence speed and stability. The **λDTMD** term serves only as a proportional weight in the early‑stopping criterion, so any positive value leads to consistent behavior; in our experiments, it was likewise fixed to 100 by default.
>
> These hyperparameters were validated on **LLaMA‑2‑7B** and then directly applied to all other model families **without re‑tuning**, demonstrating their robustness.
>
> To ensure full reproducibility, we will **release the complete training code and configuration files**, including optimizer settings, loss coefficients, and initialization parameters, upon publication.
>
> >Q3&Q7: (3) The "Related Work" … compared or analyzed. (7) The work lacks … the addition of learnability.
>
> We agree that a clearer articulation of the distinction between existing methods and our proposed framework would strengthen the contribution of our work.
>
> **(1) Different from SmoothQuant.**
>
> SmoothQuant performs static scaling factor redistribution between weights and activations to mitigate activation outliers. In contrast, our **T‑transformation** is a **learnable mapping** that adaptively reshapes each layer’s entire weight distribution from unimodal to bimodal form. This transformation is not a simple rescaling: it is optimized under **dual‑target loss functions** (DTMD, DTNP) derived from **Theorems 1 and 2**, which explicitly guide the weight distribution toward two distinct modes compatible with 1‑bit quantization. SmoothQuant cannot achieve this transition, as shown in **Table 3**, where our learnable transformation consistently outperforms static scaling even when activation quantization bit‑widths are identical.
>
> **(2) Different from QAT approaches (e.g., BitNet).**
>
> While QAT methods reshape weights into double‑bell distributions through full model retraining, they require **tens of GPU‑hours** and modify model parameters globally. In contrast, **DBellQuant achieves a similar distributional effect post‑training**, learning only a lightweight transformation T via a short optimization phase (**≈ 0.5hour on a single RTX 4090 GPU for LLaMA‑2‑7B**). This makes DBellQuant **the first PTQ framework to reach near‑QAT accuracy under 1‑bit weight and low‑bit activation quantization** with minimal computational overhead.
>
> While our design is inspired by existing concepts, its key novelty lies in **formulating the dual‑bell transformation as a learnable, theoretically grounded, and computationally efficient PTQ framework**—bridging the gap between static scaling (SmoothQuant) and resource‑intensive QAT (BitNet).
>
> >Q4:  In the method section … critical implementation step unclear.
>
> To achieve **x * T⁻¹**,  based on the Equation.3 , we simply divide the LayerNorm weights and bias (if present) by **T**, which effectively implements the desired transformation.
>
> code:
> ```
> ln.weight.div_(T)
> ln.bias.div_(T)
> ```

---

> ### Author Response · Authors · 2025-11-25
> **Response to Reviewer ETLQ(2/3)**
>
> >Q5: The paper only … support this observation.
>
> In **Section 3.4**, we provide a detailed empirical analysis showing that the inverse transformation $T^{-1}$ effectively contracts activation ranges and suppresses outliers. Here, we further supplement this with a theoretical explanation.
>
> - **Theoretical Mechanism of Activation Outlier Compression**
>
> (1) **When |Wⱼ| is relatively small**
>
> As shown in **Equation (4)**, when a channel *j* contains large activation outliers (|Xⱼ| is large) and its corresponding |Wⱼ| is relatively small (a common scenario for quasi-Gaussian weight distributions), the initialized |Tⱼ| is greater than 1. During training, as analyzed in **Theorem 2**, our **dual-target losses** (**Eqs. (5)** and **(6)**) further enlarge |Tⱼ| to better align the distribution with a bimodal shape. Consequently, the inverse scaling factor |Tⱼ⁻¹| becomes smaller than 1. This scaling attenuates activation outliers, as |Xⱼ|·|Tⱼ⁻¹| < |Xⱼ|, effectively compressing the activation range and mitigating the impact of these outliers.
>
> (2) **When |Wⱼ| is an outlier**
>
> In scenarios where both |Xⱼ| and |Wⱼ| are large, the initialized |Tⱼ| will still be greater than 1 because |Wⱼ| is generally smaller than |Xⱼ| in magnitude. During training, |Tⱼ| will decrease to align the distribution with a bimodal shape. However, **|Tⱼ| will not decrease below 1**, and this behavior can be attributed to the following two key factors:
>
> 1. **Weight distribution characteristics**: Although |Wⱼ| being an outlier indicates the presence of large values in the *j*th channel, the overall weight distribution remains approximately Gaussian, with the majority of weights being relatively small. If |Tⱼ| were to decrease below 1, these smaller weights would effectively be scaled up, moving farther away from the quantization center. This would lead to a significant increase in the quantization error for the majority of weights, outweighing any potential reduction in loss from compressing the few larger weights. As a result, |Tⱼ| is discouraged from decreasing below 1.
> 2. **Effect of the DTNP loss**: The **DTNP loss** inherently couples the scaling factor |Tⱼ| with the bimodal distribution targets (**|m₁|** and **|m₂|**). As |Tⱼ| decreases, either **|m₁|** or **|m₂|** would also decrease, which would lead to an increase in the overall loss. This coupling effectively regularizes |Tⱼ|, preventing it from decreasing excessively and ensuring that the scaling factor remains greater than 1 during training.
>
> - **Empirical validation of activation smoothing.**
>
> The final |Tⱼ| value is determined by the **optimization objective**, rather than being constrained by any **explicit analytical formulation**. Unlike SmoothQuant, our transformation is not derived from a closed-form scaling rule but is instead learned dynamically during training. As a result, its behavior reflects data-driven convergence rather than strict analytical guarantees, making it inherently difficult to explain through rigorous theoretical proofs. To address this, we provide **comprehensive empirical validation** by showcasing the results obtained from actual training, which effectively demonstrate the soundness of our approach.
>
> **（1）Appendix A.13** visualizes concrete activation ranges before and after DBellQuant. For instance, in LLaMA‑2‑7B, `q_proj` activations contract from [–3, 3] to [–0.5, 0.5], and `gate_proj` activations from [–1.5, 1.5] to [–0.5, 0.4].
>
> **（2）Appendix A.14** quantifies this effect via Z‑score and relative deviation error analysis, confirming that DBellQuant substantially reduces both absolute and relative activation outliers.
>
> Taken together, these theoretical insights and quantitative results demonstrate that the learned transformation implicitly balances both weight and activation spaces—suppressing activation outliers.

---

> ### Author Response · Authors · 2025-11-25
> **Response to Reviewer ETLQ(3/3)**
>
> >Q6: The reason why … two loss functions.
>
> **(1) Why DTMD causes shrinkage of *T*.**
>
> As explained in Section 3.3, the **DTMD** loss encourages all transformed weights (W·T) to move closer to the two quantized centers (m₁, m₂) to form a bimodal distribution. However, this formulation unintentionally introduces a degenerate solution: when *T* is scaled toward zero, both |W·T| and the corresponding m₁, m₂, computed from Eq. (2), approach zero simultaneously. This reduces the residual term |W·T – m₁| or |W·T – m₂|, the distance between the weights after transfomred and the target points becomes trivially small, driving the DTMD loss to near-zero,  causing a **loss hack** effect. Therefore it **fails** to achieve the goal of transforming the weight distribution from unimodal to bimodal. Consequently, T continues shrinking, the weight dynamic range is compressed, and quantization difficulty is **unintentionally** transferred to activations.
>
> **(2) Why DTNP is introduced.**
>
> To address this issue, we introduce the **DTNP (Dual‑Target Normalized Proportional)** loss, which is derived directly from DTMD by dividing each deviation term by |m₁| or |m₂|. This normalization constrains the update magnitude of *T*. In doing so, DTNP prevents *T* from increasing indefinitely and guides it to learn a **proper** transformation that gradually reshapes the weight distribution from unimodal to bimodal, enabling a **stable and progressive** transformation process.
>
> **(3) Why both DTMD and DTNP are necessary.**
>
> Although DTNP stabilizes training, using it alone may cause the optimization trajectory to deviate from the ideal bimodal direction. As observed in **Figure 4**, training with DTNP alone makes DTMD first decrease and then rebound—indicating over‑adaptation rather than convergence. Therefore, we combine **DTMD + DTNP** with an **early‑stopping mechanism** triggered by DTMD’s minimum point. This joint design ensures both convergence toward a clean dual‑bell distribution and avoidance of collapse. Ablation results in the below table confirm that the combined losses achieve the **most stable and accurate** quantization among all variants.
>
> | LLaMA-2-7b    | wikitext2 | C4    |
> |---------------|-----------|-------|
> | DTMD+DTNP     | 17.91     | 21.83 |
> | only DTNP     | 28.74     | 31.42 |

---

### Official Review · Reviewer_KFsn · 2025-10-21

**Soundness:** 2
**Presentation:** 3
**Contribution:** 2
**Rating:** 4
**Confidence:** 4

**Summary:**

This paper proposes DBellQuant, a post-training quantization (PTQ) framework that simultaneously achieves near 1-bit weight binarization and 6-bit activation quantization for large language models (LLMs). The core idea is to apply a learnable, per-channel transformation T to reshape the single-bell (Gaussian-like) weight distribution into a dual-bell form—arguably more amenable to binarization—while applying the inverse transformation $T^{−1}$
  to activations to preserve functional equivalence and suppress outliers. The method introduces a lightweight optimization algorithm (LTDB) with a custom dual-target loss and early stopping to learn $T$.

**Strengths:**

1) The authors conduct thorough experiments on multiple LLMs, demonstrating the effectiveness of DBellQuant across different model sizes and architectures.

2) The authors perform extensive ablation studies to validate the effectiveness of their proposed Learnable Transformation for Dual-Bell (LTDB) algorithm and the impact of various hyperparameters.

**Weaknesses:**

1) While the transformation of weight distributions into dual-bell shapes is novel, the overall framework of DBellQuant can be seen as a combination of existing techniques (e.g., learnable transformations, activation smoothing). The authors do not sufficiently justify why this specific combination leads to significantly better results compared to other potential combinations or state-of-the-art methods.

2) More directly, the paper compares against BiLLM and ARB-LLM but omits comparison with rotation-based PTQ methods (e.g., QuaRot, SpinQuant, DuQuant), which also aim to suppress outliers and enable low-bit quantization via learned or fixed orthogonal transforms. These works are cited but not evaluated.

3) The paper does not report the practical speedup or memory savings after LLM binarization using DBellQuant.

**Questions:**

How about the results of DBellQuant on reasoning large models and multi-modal large models?

---

> ### Author Response · Authors · 2025-11-25
> **Response to Reviewer KFsn**
>
> Dear Reviewer KFsn,
>
> Thank you for highlighting and appreciating our contributions in exploring the ultra-low-bit weight quantization in LLM. Below, we will address your questions one by one.
>
> >Q1: While the transformation … state-of-the-art methods.
>
> While DBellQuant indeed draws inspiration from prior work on weight transformation and activation smoothing, its **core contribution lies in the integrated, equivalence‑preserving dual‑bell transformation framework**, which unifies these techniques under a single theoretical and practical design.
>
> As shown in **Theorem 2** (with detailed proof in Appendix  A.16), bimodal weight distributions are inherently more compatible with 1‑bit binarization because they minimize quantization error relative to unimodal (Gaussian‑like) distributions. Building on this theoretical insight, we introduce a **learnable transformation T** that adaptively reshapes weight distributions toward dual‑bell forms and employ **two custom loss functions** with an early‑stopping mechanism to guarantee stable convergence without retraining. These components were not used in prior PTQ works and are explicitly tailored for near‑1‑bit quantization.
>
> Further, to maintain computational equivalence, the **inverse transformation T⁻¹** is applied to activations. This coupling is not a mere combination but a deliberate design choice: it preserves the functional output of each layer while simultaneously **suppressing activation outliers** (Section 3.4), enabling stable **6‑bit activation quantization** alongside 1‑bit weights, something that existing PTQ methods have not achieved.
>
> Empirically, our framework demonstrates **substantial improvements** over prior methods (e.g., BiLLM, ARB‑LLM) across **all model families and bit‑widths** (Tables 1–2), confirming that this specific combination is both **theoretically grounded** and **empirically validated** as essential to DBellQuant’s superior performance.
>
> >Q2:  More directly … but not evaluated.
>
> The reason that we didn’t include them for comparisons is that these approaches are not designed for 1-2 bit quantization. While these approaches such as **QuaRot**, **SpinQuant**, and **DuQuant** effectively mitigate activation outliers through orthogonal transformations, they typically experience **substantial performance drops when weights are quantized below 4 bits**. Since **DBellQuant** specifically targets **near‑1‑bit weight quantization**, we focused our comparisons on state‑of‑the‑art **binarization‑oriented** methods (e.g., BiLLM, ARB‑LLM) that operate in the same bit‑width regime.
>
> | LLaMA-2-7b   | 2bit  | 1bit |
> |--------------|-------|--------|
> | quarot+rtn   | Inf   | Inf    |
> | quarot+gptq  | 22.07 | Inf    |
> | DBellQuant 1.1bit  | - |    17.91    |
>
> |LLaMA-2-13B  | 2bit  | 1bit |
> |--------------|-------|--------|
> | quarot+rtn   | Inf   | Inf    |
> | quarot+gptq  | 10.41 | Inf    |
> | DBellQuant 1.1bit  | - |    12.79    |
>
> >Q3:  The paper does not … LLM binarization using DBellQuant.
>
> **Section 4.4** of the paper reports both **training/inference time** and **memory consumption** comparisons. Specifically, **Table 15** summarizes the memory footprint and effective bit‑width for LLaMA‑2‑7B, with additional results for other model families provided in **Appendix A.10**. Moreover, **Table 16** presents the **inference speedup** achieved by DBellQuant, and the corresponding computational details and formulas are described in **Appendix A.11**.
>
> >Q4: How about the … multi-modal large models?
>
> For multi-modal large models, we evaluated **Qwen-2.5-VL-7B** on TextVQA, ChartQA, and MME datasets, the results are shown in the below table. While performance drops significantly when quantized to near 1-bit, it still outperforms GPTQ 2-bit, where some tasks produce invalid outputs (e.g., gibberish) leading to zero scores. Our method also shows an advantage over BiLLM, **highlighting its effectiveness** in ultra-low-bit quantization.
>
> |     Qwen-2.5-VL-7B           | TextVQA | ChartQA | MME-perception |
> |----------------|---------|---------|----------------|
> | fp16           | 84.9    | 87.3    | 1695    |
> | gptq 2bit      | 0       | 0       | 319       |
> | BiLLM 1.1bit   | 26.3    | 3.7     | 638    |
> | DBellQuant 1.1bit | 28.4 | 5.2     | 742    |
>
> For the reasoning model, we used **DeepSeek-R1-Distill-Qwen-7B** and evaluated it on MathQA and LogiQA2, the results are shown in the below table. . While performance drops noticeably when quantized to near 1-bit, it still outperforms GPTQ 2-bit and BiLLM, demonstrating the **advantage** of our method in ultra-low-bit quantization.
>
> | Deepseek-R1-Distill-Qwen-7B | mathqa        | logiqa2       |
> |-------|---------------|---------------|
> | fp16                        | 44.96 ± 0.91 | 29.58 ± 1.15 |
> | gptq 2bit                       | 20.51 ± 0.82 | 14.65 ± 1.53 |
> | BiLLM    1.1bit                   | 24.99 ± 0.79 | 20.42 ± 1.87 |
> | DBellQuant      1.1bit            | 27.65 ± 0.75 | 22.78 ± 1.48 |

---

### Official Review · Reviewer_XN14 · 2025-10-28

**Soundness:** 1
**Presentation:** 2
**Contribution:** 1
**Rating:** 2
**Confidence:** 4

**Summary:**

This work connects the bimodal distribution with 1-bit quantization, which is somewhat intuitive. On this basis, a corresponding smooth coefficient is trained for each weight to form a bimodal distribution, and finally achieves SOTA results under the W1A6 setting.

**Strengths:**

This work believes that the bimodal distribution is beneficial for binarization, which is somewhat intuitive.
The writing is clear and easy to understand.

**Weaknesses:**

1.
There is no strict theoretical connection between the bimodal distribution and 1-bit quantization. For example, in extreme cases, an entire channel may belong to a single peak, which limits the rigor of this work.

2.
After applying Smooth, the quantization error of the weights cannot truly reflect the quantization error because activations also have differences between value channels.

3.
Only results on the LLaMA-7B model are reported, lacking results on larger-scale models such as 70B. This is not conducive to demonstrating the universality of the method. Additionally, reporting results on models like Qwen and Mixtral would enhance the universality of the method.

4.
The design of the training loss is also overly complex, introducing too many uncertain factors such as hyperparameter adjustments and early termination of training.

5.
The 1-bit quantization in arb-llm involves too many smooth operations, which actually require complex dequantization and cannot utilize low-precision tensor cores, often providing no help in reducing computational load. Furthermore, it lacks detailed latency statistics (especially for large-batch prefill). 6-bit quantization often fails to achieve acceleration (generally, 4/8-bit quantization is more conducive to acceleration).

**Questions:**

Activated 6-bit quantization is often detrimental to memory alignment. It can provide activated 4-bit results. Are these results usable?

---

> ### Author Response · Authors · 2025-11-25
> **Response to Reviewer XN14(1/4)**
>
> Dear Reviewer XN14,
>
> Thank you for taking the time to provide your valuable and professional suggestions on our paper. We will address each of your questions one by one.
>
> >Q1:  There is no … of this work.
>
> Indeed, extreme cases where an entire channel falls into a single peak can theoretically occur. However, our analysis and empirical evidence indicate that such cases are rare and have negligible impact on the overall quantization behavior.
>
> First, prior works (e.g., [1, 2, 3]) consistently show that **LLM weight distributions follow a quasi‑Gaussian pattern across channels**, rather than multiple isolated peaks. Consequently, most channels retain similar statistical properties, which allows our **learnable transformation T** to effectively reshape the global weight block toward a **dual‑bell configuration** that is better aligned with 1‑bit quantization levels (–1, +1). Even if a few channels remain unimodal, the optimization of  T minimizes their contribution to global quantization error, resulting in a **substantial reduction in block‑level quantization loss**, as confirmed in Table 1 and Figure 6.
>
> Second, as demonstrated in **Theorem 2 (Appendix A.16)**, we provide a mathematical justification that transforming a Gaussian‑like (single‑bell) distribution into a bimodal form inherently reduces the expected quantization error. This theoretical analysis establishes a direct connection between the **bimodal distribution property and binary quantization fidelity**, rather than relying on empirical coincidence.
>
> Finally, we emphasize that the purpose of DBellQuant is not to assume an inherent dual‑bell structure in weights but to **learn the optimal  T** that transforms the original, hard‑to‑binarize unimodal distribution into a quantization‑friendly bimodal one. Even under rare channel‑level degeneracies, this transformation adaptively rescales and redistributes weights, ensuring **robustness and consistent error reduction** across all layers.
>
> [1]Post-Training Piecewise Linear Quantization for Deep Neural Networks. ECCV 2020
>
> [2]LLM.int8(): 8-bit Matrix Multiplication for Transformers at Scale. NeurIPS 2022
>
> [3]Weight Uncertainty in Neural Networks. ICML 2015

---

> ### Author Response · Authors · 2025-11-25
> **Response to Reviewer XN14(2/4)**
>
> Q2:  After applying Smooth … between value channels.
>
> In **Section 3.4**, we provide a detailed empirical analysis showing that the inverse transformation $T^{-1}$ effectively contracts activation ranges and suppresses outliers. Here, we further supplement this with a theoretical explanation.
>
> - **Theoretical Mechanism of Activation Outlier Compression**
>
> (1) **When |Wⱼ| is relatively small**
>
> As shown in **Equation (4)**, when a channel *j* contains large activation outliers (|Xⱼ| is large) and its corresponding |Wⱼ| is relatively small (a common scenario for quasi-Gaussian weight distributions), the initialized |Tⱼ| is greater than 1. During training, as analyzed in **Theorem 2**, our **dual-target losses** (**Eqs. (5)** and **(6)**) further enlarge |Tⱼ| to better align the distribution with a bimodal shape. Consequently, the inverse scaling factor |Tⱼ⁻¹| becomes smaller than 1. This scaling attenuates activation outliers, as |Xⱼ|·|Tⱼ⁻¹| < |Xⱼ|, effectively compressing the activation range and mitigating the impact of these outliers.
>
> (2) **When |Wⱼ| is an outlier**
>
> In scenarios where both |Xⱼ| and |Wⱼ| are large, the initialized |Tⱼ| will still be greater than 1 because |Wⱼ| is generally smaller than |Xⱼ| in magnitude. During training, |Tⱼ| will decrease to align the distribution with a bimodal shape. However, **|Tⱼ| will not decrease below 1**, and this behavior can be attributed to the following two key factors:
>
> 1. **Weight distribution characteristics**: Although |Wⱼ| being an outlier indicates the presence of large values in the *j*th channel, the overall weight distribution remains approximately Gaussian, with the majority of weights being relatively small. If |Tⱼ| were to decrease below 1, these smaller weights would effectively be scaled up, moving farther away from the quantization center. This would lead to a significant increase in the quantization error for the majority of weights, outweighing any potential reduction in loss from compressing the few larger weights. As a result, |Tⱼ| is discouraged from decreasing below 1.
> 2. **Effect of the DTNP loss**: The **DTNP loss** inherently couples the scaling factor |Tⱼ| with the bimodal distribution targets (**|m₁|** and **|m₂|**). As |Tⱼ| decreases, either **|m₁|** or **|m₂|** would also decrease, which would lead to an increase in the overall loss. This coupling effectively regularizes |Tⱼ|, preventing it from decreasing excessively and ensuring that the scaling factor remains greater than 1 during training.
>
> - **Empirical validation of activation smoothing.**
>
> The final |Tⱼ| value is determined by the **optimization objective**, rather than being constrained by any **explicit analytical formulation**. Unlike SmoothQuant, our transformation is not derived from a closed-form scaling rule but is instead learned dynamically during training. As a result, its behavior reflects data-driven convergence rather than strict analytical guarantees, making it inherently difficult to explain through rigorous theoretical proofs. To address this, we provide **comprehensive empirical validation** by showcasing the results obtained from actual training, which effectively demonstrate the soundness of our approach.
>
> **（1）Appendix A.13** visualizes concrete activation ranges before and after DBellQuant. For instance, in LLaMA‑2‑7B, `q_proj` activations contract from [–3, 3] to [–0.5, 0.5], and `gate_proj` activations from [–1.5, 1.5] to [–0.5, 0.4].
>
> **（2）Appendix A.14** quantifies this effect via Z‑score and relative deviation error analysis, confirming that DBellQuant substantially reduces both absolute and relative activation outliers.
>
> Taken together, these theoretical insights and quantitative results demonstrate that the learned transformation implicitly balances both weight and activation spaces—suppressing activation outliers.

---

> ### Author Response · Authors · 2025-11-25
> **Response to Reviewer XN14(3/4)**
>
> >Q3: Only results on … of the method.
>
> We need to clarify that **Table 1** includes the performance of larger-scale models such as **LLaMA-2-13B and LLaMA-2-70B**. Additionally, we have provided results for LLaMA-1-65B to further support our findings.
>
> | LLaMA-1-65B | wikitext2 | c4    |
> |-------------|-----------|-------|
> | fp16        | 3.53      | 25.07 |
> | BiLLM       | 8.37      | 44.68 |
> | ARB-LLM     | 7.27      | 36.08 |
> | DBellQuant  | 6.92      | 34.52 |
>
> **Qwen family tests:** To further assess generalization, we additionally implemented **DBellQuant** on the **Qwen‑2‑7B** and **Qwen‑2.5‑7B** models, a recently released architecture known for its strong performance. The new results, now included in the revised paper (see Table  9 and 10), demonstrate consistent improvements over baseline PTQ methods, suggesting that **DBellQuant generalizes well across different LLM architectures** without requiring model‑specific modifications.
>
> | Qwen-2-7B  | wikitext2 | C4    |
> |------------|-----------|-------|
> | BiLLM      | 38.42     | 40.81 |
> | DBellQuant | 30.47     | 35.02 |
>
> | Qwen-2.5-7B | wikitext2 | C4    |
> |-------------|-----------|-------|
> | BiLLM       | 41.74     | 53.07 |
> | DBellQuant  | 33.47     | 43.18 |
>
> >Q4:  The design of … termination of training.
>
> Our loss design is intentionally minimal yet **theoretically grounded** to ensure both convergence stability and interpretability.
>
> Building on **Theorem 2**, which formally connects bimodal weight distributions with reduced quantization error, we introduce two complementary objectives—**Dual‑Target Minimum Deviation (DTMD)** and **Dual‑Target Normalized Proportional (DTNP)** losses—to guide the transformation *T* toward an optimal dual‑bell distribution. The design is conceptually simple: **DTMD** encourages weights to cluster near two target means, while **DTNP** normalizes this process to prevent degeneracy. Together, they provide a direct and monotonic optimization path supported by the theory derived in **Appendix A.16**.
>
> Regarding the reviewer’s concern about hyperparameter sensitivity and early stopping:
>
> - **Appendix A.4 and A.5** detail the full training algorithm, early‑termination criterion, and default parameters.
> - The early‑stop rule is deterministic, which is triggered when DTMD ceases to decrease. Thus, it requires no manual tuning.
> - Hyperparameters such as loss coefficients and learning rates were found robust and transferable across model families (OPT, LLaMA) without re‑tuning.
>
> Empirically, as shown in **Tables 1 and 2**, the training consistently converges smoothly across all tested models, demonstrating that the proposed losses introduce minimal uncertainty and yield stable, reproducible optimization behavior.
>
> >Q5: The 1-bit quantization in arb-llm … conducive to acceleration.
>
> Indeed, **ARB‑LLM** introduces multiple smooth operations and dequantization steps, which hinder efficient tensor‑core utilization and ultimately diminish the expected acceleration benefits.
>
> In contrast, **DBellQuant** avoids these pitfalls by **not relying on complex smooth functions**. Instead, we employ a **lightweight learnable transformation *T*** that directly adjusts the weight distribution toward a dual‑bell shape, thereby reducing quantization error without introducing additional nonlinear transformations or computational overhead. This design ensures **full compatibility with standard low‑precision integer operators (INT8)** commonly supported by hardware accelerators.
>
> **Latency statistics:** We also provide detailed latency measurements in **Table 16**(also can be seen below) and computational breakdowns in **Appendix A.11**. The results show that DBellQuant achieves consistently **lower inference latency** than ARB‑LLM and BiLLM across representative linear layers of LLaMA‑2‑7B, confirming that our approach can **leverage INT8 tensor cores for activation computation** while maintaining near‑1‑bit weight precision.
>
> | Model       | 4096×4096 | 4096×11008 | 11008×4096 |
> |-------------|-----------|------------|------------|
> | FP16        | 0.76595   | 1.63532    | 1.76949    |
> | PB-LLM      | 0.73363   | 1.44076    | 1.69881    |
> | BiLLM       | 0.34201   | 0.36777    | 0.37689    |
> | ARB-LLM-X   | 0.33180   | 0.35539    | 0.36792    |
> | DBellQuant  | 0.27694   | 0.30085    | 0.31860    |

---

> > ### Author Response · Authors · 2025-11-25
> > **Response to Reviewer XN14(4/4)**
> >
> > >Q6:  Activated 6-bit … these results usable?
> >
> > - **Failure below 6-bit activations:**
> >
> >     As noted, quantizing activations to 4 bits remains an open challenge. The primary reason lies in **information bottleneck compounding**: our framework already binarizes weights (≈1 bit), resulting in **substantial representational compression**. Further reducing activations below 6 bits leads to **excessive information loss** in intermediate representations, particularly in attention and feed-forward modules, which are highly sensitive to activation magnitude distributions. Even though DBellQuant’s dual-bell transformation successfully regularizes activations and enables stable 6-bit quantization, pushing to 4 bits causes **severe quantization noise accumulation**, as the reduced dynamic range fails to capture key semantic variations.
> >
> > - **Empirical evidence at 4 bits:**
> >
> >     We conducted additional experiments on **LLaMA‑2‑7B**. Compared to the baseline BiLLM, DBellQuant achieved perplexity reductions on **Wikitext2** (2637 → 271) and **C4** (3502 → 232), a **~90% improvement**, indicating substantial robustness. However, the model still fails to generate coherent text, confirming that while DBellQuant mitigates degradation, **4-bit activations remain below the usable fidelity threshold** for stable language modeling. This is still a promising future direction to enable models to function effectively under 4-bit activation quantization with binary weight.
> >
> > | Llama-2-7b           | wikitext2 | c4   |
> > |-----------------------|-----------|------|
> > | fp16                 | 5.47      | 7.26 |
> > | BiLLM (w1a4)         | 2637      | 3502 |
> > | DBellQuant (w1a4)    | 271       | 232  |

---

### Official Review · Reviewer_TfoS · 2025-11-01

**Soundness:** 3
**Presentation:** 2
**Contribution:** 3
**Rating:** 6
**Confidence:** 3

**Summary:**

This paper introduces DBellQuant, a post-training quantization (PTQ) framework targeting highly compressed large language models by applying simultaneous near-1-bit weight binarization and 6-bit activation quantization with minimal performance loss. The central innovation is the Learnable Transformation for Dual-Bell (LTDB) algorithm, which transforms unimodal (single-bell) weight distributions into dual-bell forms, making weights more amenable to binarization while inversely transforming activations to reduce outlier effects and facilitate low-bit quantization. The framework is empirically shown to outperform state-of-the-art PTQ methods, such as BiLLM and ARB-LLM, on several LLM families and benchmarks, preserving accuracy and language modeling ability under aggressive quantization.
This paper presents a valuable contribution to LLM post-training quantization through the DBellQuant framework, whose core LTDB algorithm reshapes weights into dual-bell distributions to enable high-compression quantization (near-1-bit weights, 6-bit activations) with minimal performance loss, outperforming SOTA methods like BiLLM and ARB-LLM. However, it suffers from relatively modest overall novelty, an unresolved 4-bit activation limitation, and several presentation and clarity issues. Nonetheless, the approach demonstrates genuine promise for efficient LLM deployment and could serve as a solid foundation for future work.

**Strengths:**

(1) Principled Foundation and Theoretical Motivation: The authors offer a thorough mathematical argument for why dual-bell weight distributions are better suited for binarization. This is explicitly elaborated in Section 3.1 and Appendix A.16, where they present formal proofs that demonstrate the soundness of the transformation and clarify its impact on the target weight distributions.
(2) Solid Empirical Validation: Experiments span multiple LLMs (LLaMA, OPT families, etc.) and metrics (perplexity on WikiText2 and C4; QA accuracy on ARC, PIQA, etc.). Quantitative results consistently demonstrate that DBellQuant outperforms strong post-training quantization baselines such as BiLLM and ARB-LLM. For instance, Table 1 and Table 2 directly show measurable improvements, especially at aggressive quantization levels.
(3) Ablation and Robustness Studies: The impact of block size, activation bits, and loss design choices are all explored systematically (Tables and commentary in ablation section). These experiments reinforce the reliability of the findings and offer valuable insights into the design choices.

**Weaknesses:**

(1) Limited Novelty Relative to Prior PTQ/Quantization Methods: Although the transition from single- to double-bell distributions is well-motivated and the learnable transformation is elegant, many individual components (e.g., smooth-scaling, block-wise transformations, loss design) are iterative extensions or combinations of well-established quantization techniques.
(2) Insufficient Depth in Activation Quantization Analysis: While it claims that the inverse transformation effectively mitigates activation outliers--supported by visual illustrations (Figures 2, 5) and perplexity improvements--it falls short of providing rigorous statistical or theoretical validation, such as quantifying reductions in kurtosis, and omits direct comparisons of activation distributions before and after smoothing. Moreover, the method is currently limited to 6-bit activations, with model collapse at 4 bits, undermining its claim to state-of-the-art status in extreme compression; this inability to support 4-bit activations represents a barrier for edge deployment scenarios.
(3) Limited Discussion of Learned Transformation Properties: While Theorem 1 (Section 3.2) establishes existence, the main text offers insufficient insight into the practical challenges of learning such transformations in high-dimensional settings, especially with constraints on invertibility and computational overhead.
(4) Editing Issues: The manuscript contains several typographical errors (e.g., “doubel-bell,” “algrithm”) and occasional awkward or imprecise phrasing that detracts from readability and professionalism.

**Questions:**

(1) Could the authors provide more analysis on why DBellQuant fails when pushing activation quantization below 6 bits? Additionally, since the current implementation only supports INT8 activation quantization, it would be valuable to explore whether integrating emerging low-precision floating-point formats (e.g., MXFP6, MXFP4, or NVFP4), which are designed to maintain representational capacity at very low bitwidths while supporting efficient hardware execution, could mitigate these limitations.
(2) Did the authors explore combining DBellQuant with other methods that promote incoherence or regularization (e.g., Hadamard-based transforms, SpinQuant and Atom) and how would these interact?
(3) Regarding outlier robustness: does LTDB effectively handle strongly outlying weights or highly non-Gaussian channels, or does it implicitly assume that the input weights are already approximately Gaussian?
(4) Could the authors offer additional discussion on how well DBellQuant generalizes to LLM architectures beyond OPT and LLaMA, or are there theoretical constraints limiting its generality?

---

> ### Author Response · Authors · 2025-11-25
> **Response to Reviewer TfoS(1/3)**
>
> Dear Reviewer TfoS,
>
> Thank you for taking the time to provide your valuable and professional suggestions on our paper. We will address each of your questions one by one.
>
> >Q1:(1) Limited Novelty Relative … quantization techniques.
>
> While DBellQuant indeed builds upon prior quantization foundations, its novelty lies in the learnable equivalence-preserving dual-bell transformation framework, which systematically connects weight distribution shaping with activation smoothing in post-training quantization.
>
> Specifically, **Theorem 1** provides a theoretical foundation demonstrating that mapping unimodal  weight distributions into bimodal forms inherently reduces binarization error, an insight not explored in previous PTQ studies. Building on this, we introduce **a learnable transformation matrix T** and two **new dual-target loss functions** that jointly regulate the shaping process, ensuring convergence toward a desired dual-bell distribution without retraining the full model. These losses (DTMD and DTNP) are explicitly formulated to maintain stability and directionality during optimization—addressing known shortcomings of earlier heuristics such as simple scaling or normalization-based redistribution.
>
> In addition, our **activation-aware initialization** and **smooth-scaling strategy** serve not merely as minor adaptations but as integral components of the equivalence transformation framework: they guarantee that the inverse of T smooths activations and makes low-bit (e.g., 6-bit) activation quantization feasible under 1-bit weight compression for the first time in a PTQ setting.
>
> Thus, while DBellQuant draws upon established quantization concepts, its contribution lies in unifying them under a theoretically grounded, learnable dual-bell transformation paradigm that empirically and analytically advances the frontier of post-training binary LLM quantization.
>
> >Q2&Q5: (2)Insufficient Depth in … edge deployment scenarios. (5)Could the authors … mitigate these limitations.
>
> - **Activation quantization analysis and statistical validation:**
>
>     We have extensively analyzed the mitigation of activation outliers in **Section 3.4**, supported by **Figures 2 and 5** showing the distributional contraction and by **Appendices A.13 and A.14** where we present both visualization of activation values and quantitative comparisons .
>
> （1）Appendix A.13 visualizes concrete activation ranges before and after DBellQuant. For instance, in LLaMA‑2‑7B, `q_proj` activations contract from [–3, 3] to [–0.5, 0.5], and `gate_proj` activations from [–1.5, 1.5] to [–0.5, 0.4].
>
> （2）Appendix A.14 quantifies this effect via Z‑score and relative deviation error analysis, confirming that DBellQuant **substantially reduces** both absolute and relative **activation outliers**.
>
> - **Failure below 6-bit activations:**
>
>     As noted, quantizing activations to 4 bits remains an open challenge. The primary reason lies in **information bottleneck compounding**: our framework already binarizes weights (≈1 bit), resulting in **substantial representational compression**. Further reducing activations below 6 bits leads to **excessive information loss** in intermediate representations, particularly in attention and feed-forward modules, which are highly sensitive to activation magnitude distributions. Even though DBellQuant’s dual-bell transformation successfully regularizes activations and enables stable 6-bit quantization, pushing to 4 bits causes **severe quantization noise accumulation**, as the reduced dynamic range fails to capture key semantic variations.
>
> - **Empirical evidence at 4 bits:**
>
>     We conducted additional experiments on **LLaMA‑2‑7B** in the table below. Compared to the baseline BiLLM, DBellQuant achieved perplexity reductions on **Wikitext2** (2637 → 271) and **C4** (3502 → 232), a **~90% improvement**, indicating substantial robustness. However, the model still fails to generate coherent text, confirming that while DBellQuant mitigates degradation, **4-bit activations remain below the usable fidelity threshold** for stable language modeling. This is still a promising future direction to enable models to function effectively under 4-bit activation quantization with binary weight.
>
> | LLama-2-7b           | wikitext2 | c4   |
> |-----------------------|-----------|------|
> | fp16                 | 5.47      | 7.26 |
> | BiLLM (w1a4)         | 2637      | 3502 |
> | DBellQuant (w1a4)    | 271       | 232  |

---

> ### Author Response · Authors · 2025-11-25
> **Response to Reviewer TfoS(2/3)**
>
> >Q2&Q5: (2)Insufficient Depth in … edge deployment scenarios. (5)Could the authors … mitigate these limitations.
>
> - **On integrating floating‑point formats (MXFP6/FP4):**
>
>     We agree that emerging low‑precision floating‑point representations (e.g., MXFP6, MXFP4, NVFP4) may retain more information at similar bitwidths by preserving exponent scaling. Our method is theoretically compatible with such formats: the learnable dual-bell transformation could be applied in the mantissa spac*, and the inverse transformation could still perform range contraction before FP quantization. However, our current implementation focuses on **integer arithmetic** to maximize hardware compatibility (e.g., INT8 kernels on GPUs and NPUs). We plan to explore hybrid integer–floating extensions in future work, which could further reduce quantization loss under extreme low-bit settings.
>
> >Q3: Limited Discussion of … computational overhead.
>
> We agree that learning an equivalent transformation in high‑dimensional settings poses several theoretical and practical challenges chiefly related to (1) the dimensionality and potential non‑invertibility of  T, (2) optimization stability, and (3) computational overhead in training and inference.
>
> To address these concerns:
>
> 1. **Dimensionality and Learnability:**
>
>     Rather than learning a full *m × m* matrix (which would be computationally prohibitive for LLMs), our formulation constrains *T* to a lightweight 1 × *C* vector applied in an elementwise fashion. This drastically reduces learning complexity while retaining sufficient flexibility to reshape each layer’s weight distribution.
>
> 2. **Invertibility and Stability:**
>
>     The equivalence‑preserving design ensures numerical stability without explicitly enforcing strict invertibility. The combination of **dual‑target loss functions (DTMD and DTNP)** and an **early‑stopping criterion** prevents overfitting or degeneracy of T. Empirically, we observe smooth and monotonic convergence across layers.
>
> 3. **Computational Overhead in training and inference**
>
>     As detailed in **Section 4.4** and **Appendix A.9**, training T introduces negligible cost. For example, in LLaMA‑2‑7B, DBellQuant requires only **12 additional minutes** on an A100 GPU (Table 6) compared to BiLLM, while achieving a ~45% perplexity reduction. This confirms that the transformation learning process is efficient and scalable. Besides, We would like to clarify that **our inference process does not introduce any additional overhead**. Specifically, the proposed method operates within the same computational complexity as the baseline approaches during inference.
>
> >Q4: Editing Issues … professionalism.
>
> Thank you for pointing this out. We acknowledge the typo and will ensure it is corrected in future revisions of the manuscript.
>
> >Q6:  Did the authors … would these interact?
>
> We explored combining **DBellQuant** with **Hadamard‑based rotation transforms (QuaRot)** on the LLaMA‑2‑7B model; however, the results were slightly worse than using **DBellQuant** alone. A likely reason is that DBellQuant explicitly reshapes the weight distribution into a **bimodal form** optimized for binarization, while the orthogonal rotations introduced by Hadamard‑based methods randomly redistribute weight values across channels. This operation disrupts the learned bimodal structure and weakens the quantization alignment achieved by **T‑transformation**.
>
> | method              | wikitext2 | C4    |
> |---------------------|-----------|-------|
> | DBellQuant          | 17.91     | 21.83 |
> | DBellQuant+Quarot   | 62.37     | 76.59 |
>
> >Q7: Regarding outlier robustness … already approximately Gaussian?
>
> In our work, we **assume that pretrained weight distributions are approximately Gaussian‑like**, which is consistent with extensive prior findings. For example, [1] reports that “the distributions of weights and activations of pre‑trained DNNs are bell‑shaped, such as Gaussian or Laplacian,” and subsequent studies [2, 3] show that **outliers constitute only a small fraction** of the overall parameters, with most values concentrated near zero.
>
> Under this assumption, **LTDB** operates effectively by reshaping the near‑Gaussian weight distribution into a bimodal form for 1‑bit quantization. Although the method is not explicitly designed for severely non‑Gaussian distributions, empirical results in diverse model families (e.g., LLaMA‑2, Qwen 2) suggest that it remains robust in typical pretrained settings where such Gaussian‑like characteristics hold.
>
> [1]Post-Training Piecewise Linear Quantization for Deep Neural Networks. ECCV 2020
>
> [2]LLM.int8(): 8-bit Matrix Multiplication for Transformers at Scale. NeurIPS 2022
>
> [3]Weight Uncertainty in Neural Networks. ICML 2015

---

> > ### Author Response · Authors · 2025-11-25
> > **Response to Reviewer TfoS(3/3)**
> >
> > >Q8: Could the authors … limiting its generality?
> >
> > Similar to prior quantization studies (e.g., OmniQuant, QuaRot), our main experiments focused on the **OPT** and **LLaMA** families, which are widely adopted benchmarks for LLM quantization research. Following this convention ensures fair and direct comparison with existing methods.
> >
> > To further assess generalization, we additionally implemented **DBellQuant** on the **Qwen‑2‑7B** and **Qwen‑2.5‑7B** models, a recently released architecture known for its strong performance. The new results, now included in the revised paper (see Table  9 and 10), demonstrate consistent improvements over baseline PTQ methods, suggesting that **DBellQuant generalizes well across different LLM architectures** without requiring model‑specific modifications.
> >
> > | Qwen-2-7B  | wikitext2 | C4    |
> > |------------|-----------|-------|
> > | BiLLM      | 38.42     | 40.81 |
> > | DBellQuant | 30.47     | 35.02 |
> >
> > | Qwen-2.5-7B | wikitext2 | C4    |
> > |-------------|-----------|-------|
> > | BiLLM       | 41.74     | 53.07 |
> > | DBellQuant  | 33.47     | 43.18 |

---

> > > ### Comment · Reviewer_TfoS · 2025-11-28
> > > **Response to Author**
> > >
> > > We appreciate the authors' detailed point-by-point responses, which have certainly clarified several aspects of our review. After considering the explanations from the perspectives of both novelty and practical utility, we have decided to maintain our original score. Thanks!

---

> > > > ### Author Response · Authors · 2025-11-28
> > > > **Response to Reviewer TfoS**
> > > >
> > > > Dear Reviewer TfoS,
> > > >
> > > > We are delighted that our responses and experiments addressed your questions and concerns, and we sincerely thank you for your suggestions!
> > > >
> > > > Best regards,
> > > >
> > > > Paper 11987 Authors

---

### Author Response · Authors · 2025-11-25
**General Response to Reviewers and ACs**

Dear Reviewers and ACs:

We sincerely thank the reviewers for their constructive comments and insightful feedback, which have greatly helped us improve the quality of our manuscript. We have carefully addressed all suggestions and made detailed revisions to the previous draft, with the main changes highlighted in blue.

Specifically, we have made the following improvement:

1. Clarify the effectiveness of jointly using DTMD and DTNP Loss and provide the ablation study results.
2. Add discussion on comparison of DBellQuant with Rotation-based method, and also the results of combining these two methods together.
3. Add implementation of our method on Qwen family models to confirm its strong generalization capability across different architectures.
4. Extend DBellQuant to reasoning and multi‑modal models, demonstrating its effectiveness in preserving both reasoning accuracy and multimodal alignment performance.

We again sincerely thank all the reviews and ACs’ effort on our submission!

Best regards,

Paper 11987 Authors

---

### Author Response · Authors · 2025-12-01
**Summary of factual errors in review**

Dear Area Chair,

We sincerely appreciate the valuable time and effort invested by all the reviewers and the AC in our manuscript. Since there were **plenty of factual errors** in the review, we will explain these one by one.

**Reviewer XN14**

Initial score is **2**. However, we found **numerous factual errors** in the review, and **the reviewer did not respond to our rebuttal during the discussion period**. Therefore, we will explain our responses to you point by point.

**1.The reviewer states that "Only results on the LLaMA-7B model are reported, lacking results on larger-scale models such as 70B."**

This statement is **incorrect**. In our paper, we clearly report results on **LLaMA-2-70B and LLaMA-2-13B** models in **Table 1**.

**2.The reviewer states that "introducing too many uncertain factors such as hyperparameter adjustments and early termination of training."**

We believe this concern has been **adequately addressed** in our paper. In **Appendix A.16**, we provide **detailed settings for all hyperparameters**, ensuring full transparency. Furthermore, **Sections 3.2 and 3.3** explain the rationale behind the design of the loss functions in detail, which eliminates uncertainty and clarifies the consistency of our training process.

**3.The reviewer states that "The 1-bit quantization in arb-llm involves too many smooth operations."**

We believe this comment is a **misunderstanding**. arb-llm is a paper accepted at ICLR 2025, which we used as a **baseline** for comparison. However, we **did not follow the methodology or approach** of arb-llm in our work, so it is **unclear why this was raised as a weakness of our paper**.

**4.The reviewer states that "it lacks detailed latency statistics."**

This is **incorrect**, as we provide **detailed** latency data and the calculation process in **Table 16 and Appendix 11** of the paper.

**5.The reviewer states that "There is no strict theoretical connection between the bimodal distribution and 1-bit quantization."**

However, in our paper, **Theorem 2 (Proof in Appendix A.16)** explicitly discusses and rigorously proves the theoretical connection between bimodal distribution and 1-bit quantization. Moreover, the training strategy proposed in our work is directly derived from this theorem to reduce quantization error. Unfortunately, it seems the reviewer **overlooked** this critical part of our paper.

**6.The reviewer states that "the quantization error of the weights cannot truly reflect the quantization error because activations also have differences between value channels."**

We believe this concern has been **addressed** in our paper. In **Section 3.4**, we provide a **detailed empirical analysis** on this topic, and in **Appendix A.13 and A.14**, we present evidence showing that DBellQuant substantially **reduces both absolute and relative activation outliers**.

**Reviewer KFsn**

Initial score is **4**. There are **factual errors** in the review, and **the reviewer did not respond to our rebuttal during the discussion period.**

**The reviewer states that “The paper does not report the practical speedup or memory savings after LLM binarization using DBellQuant.”**

This statement is **incorrect**. **Section 4.4** of the paper provides **detailed comparisons of both training/inference time and memory consumption**. Specifically, **Table 15** reports the **memory footprint and effective bit width** for LLaMA-2-7B, with additional results for other model families included in **Appendix A.10**. Furthermore, **Table 16** presents the **inference speedup** achieved by DBellQuant, with the corresponding computational details and formulas described in **Appendix A.11**.

**Reviewer ELTQ**

Initial score is **4**. There are **factual errors** in the review, and **the reviewer did not respond to our rebuttal during the discussion period.**

**The reviewer states that “No formulas or schematic diagrams are provided to illustrate the integration details.”**

This statement is **incorrect**.  We have introduced **Equation.3** and the **implementation is based on the Equation.3**. We simply divide the LayerNorm weights and bias (if present) by T.

We sincerely thank the Area Chair for your efforts, especially during this challenging period when the workload is particularly heavy. However, we would like to bring to your attention some critical concerns regarding our paper. We believe that our work has received **very low-quality reviews filled with factual errors**, leading to unfair treatment and an unjust score. Additionally, **we did not receive any constructive engagement during the discussion period to address these issues**. As one of the leading conferences in AI, we hold ICLR in the highest regard and hope to receive a **fair and objective evaluation** of our work. Once again, we deeply appreciate your time and effort. Thank you!

Best Regards，

Paper 11987 Authors

---

### Author Response · Authors · 2025-12-01
**Summary of rebuttal**

Dear AC,

We sincerely thank you for your time. We summarize the key point below.



| Reviewer | Strength | Questions | Initial rate | Our reply | Feedback |
|---------|----------|-----------|--------------|------------|----------|
| TfoS | Principled foundation and theoretical motivation, solid empirical validation, ablation and robustness studies. | **A.** Analysis of performance degradation under 4-bit activation quantization condition. **B.** Combination of DBellQuant with other methods. **C.** Outlier robustness validation. **D.** Generality of the method.| 6| **1.** We provided 4-bit quantization results with an explanation of the performance degradation **2.** We conducted experiments combining our method with other state-of-the-art approaches **3.** We use **extensive findings in prior work** to validate our assumptions. **4.** We showed the results on Qwen Family models.| Overall satisfied |
| XN14 | Idea is intuitive. The writing is clear and easy to understand | **A.** Results on large scale models and different models. **B.** Uncertain factors. **C.** Computation speed and latency statistics. **D.** 4 bit activation results.| 2 | **Report factual errors, details can be seen in below factual errors summary.** **1.** Report LLaMA-2-70B in Table 1, provide Qwen results. **2.** All parameters and algorithms shown in Appendix A.16 **3.** No relationship with arb-llm and latency data in Table 16. **4.** We provided 4-bit quantization results with an explanation of the performance degradation | Without Reply |
| KFsn | Thorough experiments on multiple LLMs, extensive ablation studies to validate the effectiveness of method. | **A.** The reason of combination of proposed methods. **B.** Comparison with rotation-based PTQ methods **C.** Practical speedup and memory savings. **D.** Results on reasoning large models and multi-modal large models.| 4 | **Report factual errors, details can be seen in below factual errors summary.** **1.** We provided a detailed explanation of the overall design and methodology. **2.** We supplemented our work with  comparison to rotation-based PTQ methods **3.** Time and memory saving data can be seen in Section 4.4. **4.** We included experimental results on on **large reasoning models and large multi-modal models**.| Without Reply |
| ELTQ | Achieve near 1-bit weight quantization combined with 6-bit activation quantization for the first time, with minimal performance loss. Low memory usage and low latency. | **A.** Missing hyperparameters. **B.** Differences between PTQ methods and QAT methods. **C.** Implementation of the T-matrix integrated into LayerNorm. **D.** Theoretical explanation of the causal relationship in activation outlier compression. **E.** Reason of DTMD loss causes T-matrix shrinkage.| 4 | **Report factual errors, details can be seen in below factual errors summary.** **1.** We provided **detailed** hyperparameters and explanations, and we commit to releasing our code for reproducibility. **2.** We **explained the differences** between our method and PTQ methods like SmoothQuant, as well as QAT methods like BitNet. **3.** We provided a detailed explanation along with the **corresponding code** for the implementation. **4.** We supplemented our paper with a **theoretical explanation** of the causal relationship in activation outlier compression. **5.** We **analyzed** why the DTMD loss causes T-matrix shrinkage and included the **requested ablation study results**.| Without Reply |

Best Regards,

Paper 11987 Authors

---

> ### Comment · Area_Chair_mBCp · 2025-12-03
>
> Thank you for summarizing the review comments and the rebuttal. I will review everything carefully.
> Your new AC

---

### Meta-Review · Area_Chair_mBCp · 2026-01-01

**Summary:**

Across the reviews, the main concerns center on theoretical justification, novelty, clarity of implementation details, experimental breadth, and practical deployment relevance. Some reviewers questioned whether the theoretical connection between dual-bell distributions and 1-bit quantization is fully rigorous, noting that certain reasoning appears intuitive and would benefit from deeper analysis. There were also doubts about the degree of innovation, as the transformation approach may resemble established techniques such as scaling-based PTQ (e.g., SmoothQuant) or QAT-style weight shaping, raising uncertainty about whether the overall framework represents a substantial methodological advance.

Reproducibility concerns were raised due to missing algorithmic details such as optimizer settings, learning rate policies, and precise values of loss coefficients. The explanation of how the T-matrix integrates with LayerNorm was seen as incomplete, and several reviewers remarked that the mechanism by which the inverse transformation suppresses activation outliers is not sufficiently explained from a causal or theoretical standpoint.

Regarding experiments, reviewers noted the absence of comparisons with certain rotation-based PTQ methods that also address outlier suppression. Some argued that evaluation should extend more broadly across model scales and modalities, including reasoning-focused architectures. Additionally, several reviewers specifically asked for stronger evidence of real deployment benefits, including detailed actual hardware latency, tensor core utilization, and memory-throughput-aware analysis of why W1A6 is preferable over other bit-width configurations. They expressed interest in understanding whether the proposed method achieves meaningful acceleration rather than only compressed representation.

Overall, while reviewers acknowledged that achieving near-1-bit weights with 6-bit activations in PTQ is an impactful goal for practical LLM deployment, they emphasized the need for clearer theoretical grounding, more comprehensive experimental comparisons, full reproducibility of the method, and more convincing hardware-validated performance claims.

**Reviewer Concerns:**

The authors’ rebuttal addresses several of the reviewers’ concerns in a convincing way, particularly around theory and reproducibility. Reviewer criticisms that the method lacked a precise theoretical connection between dual-bell distributions and 1-bit quantization are largely answered by the clarification of Theorem 2 and the additional explanation of how the training strategy is derived from that result. Likewise, concerns about missing implementation details are substantially mitigated: the authors now specify optimizer choice, learning rate, loss coefficients, and the concrete way the T-transformation is integrated into LayerNorm, and they commit to releasing code. Some of the alleged factual issues in the reviews, such as the absence of large-model results or latency and memory measurements, are indeed resolved by pointing to existing tables and appendices in the submission. Overall, I think the rebuttal makes it clear that the dual-bell shaping idea is nontrivial and interesting in the 1-bit setting, and that the core LTDB mechanism is better grounded and more reproducible than the initial reviews suggested.

At the same time, I believe some important concerns remain outstanding, especially those related to activation quantization and practical efficiency. Multiple reviewers either implicitly or explicitly raised that activation quantization is now a central theme in modern LLM compression, with growing attention to outlier handling and low-precision formats such as MXFP4. I share this concern. While the paper demonstrates impressive progress on near-1-bit weight quantization, the method still depends on 6-bit activation quantization, and the 4-bit activation setting is unstable or unusable. Given how critical activation precision is for end-to-end speedups and memory savings, this limitation affects the practical benefits of the approach. In particular, despite strong results under the W1A6 configuration compared to existing PTQ work, the overall performance across diverse LLM tasks still remains notably below higher-precision baselines, suggesting that additional advances on the activation side will be required before such models can be widely deployed in real-world settings.

In summary, I think the rebuttal satisfactorily addresses many of the reviewers’ points about theoretical justification, clarity of the method, and factual omissions in the original reviews. However, I also agree with the spirit of the remaining concerns around activation quantization: the current work is still heavily weight-centric, and without a robust solution for lower-bit activations, especially 4-bit, the full benefits of near–1-bit PTQ are not yet realized.

**Reviewer Scores:**

Based on the rebuttal exchange and the issues that were clarified, I believe some reviewers would likely have increased their scores slightly had they fully engaged in the discussion. In particular, several critical comments stemmed from factual misunderstandings about the existence of large-model results, latency evaluations, and theoretical justification, all of which were addressed directly and thoroughly in the rebuttal. For those reviewers, the clarified information may reasonably have improved their assessment regarding soundness and clarity.

However, despite those improvements, I think it is unlikely that the overall evaluation would have shifted dramatically. The remaining concerns related to activation quantization and practical deployability are substantial and were consistently emphasized across multiple reviews. While the contribution toward near-1-bit weight quantization is strong and well supported, the continued reliance on 6-bit activations and the inability to maintain performance under lower activation precision (e.g., 4-bit) represent a fundamental limitation in terms of real-world efficiency. These points reflect core expectations in current LLM compression literature and cannot be fully resolved within the scope of the rebuttal.

Therefore, although some upward adjustment in scores could reasonably be expected had reviewers fully engaged with the rebuttal, especially in terms of correcting factual misunderstandings, the presence of these deeper limitations suggests that any improvements would likely be incremental rather than significant.

---

### Decision · Program_Chairs · 2026-01-26

Reject